# LoCoT2V-Bench: A Benchmark for Long-Form and Complex Text-to-Video Generation

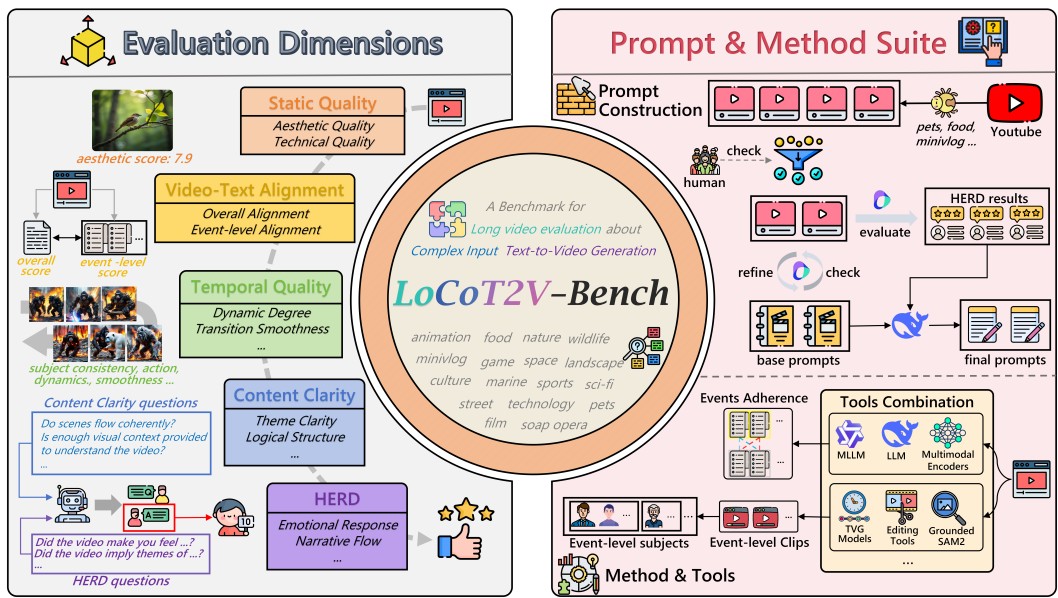

Figure 1: Overview of the **LoCoT2V-Bench**. **LoCoT2V-Bench** comprehensively evaluates the generated long videos from five dimensions: static quality, text-video alignment, temporal quality, content clarity and Human Expectation Realization Degree (HERD). We obtain our prompts from collected real-world videos via MLLMs and leverage multiple tools to execute our assessment.

## ABSTRACT

Recently text-to-video generation has made impressive progress in producing short, high-quality clips, but evaluating long-form outputs remains a major challenge especially when processing complex prompts. Existing benchmarks mostly rely on simplified prompts and focus on low-level metrics, overlooking fine-grained alignment with prompts and abstract dimensions such as narrative coherence and thematic expression. To address these gaps, we propose LoCoT2V-Bench, a benchmark specifically designed for long video generation (LVG) under complex input conditions. Based on various real-world videos, LoCoT2V-Bench introduces a suite of realistic and complex prompts incorporating elements like scene transitions and event dynamics. Moreover, it constructs a multi-dimensional evaluation framework that includes our newly proposed metrics such as event-level alignment, fine-grained temporal consistency, content clarity, and the Human Expectation Realization Degree (HERD) that focuses on more abstract attributes like narrative flow, emotional response, and character development. Using this framework, we conduct a comprehensive evaluation of nine representative LVG models, finding that while current methods perform well on basic visual and temporal aspects, they struggle with inter-event consistency, fine-grained alignment, and high-level thematic adherence, etc. Overall, LoCoT2V-Bench provides a comprehensive and reliable platform for evaluating long-form complex text-to-video generation and highlights critical directions for future method improvement.

Table 1: Comparison of benchmarks in terms of sample scale, average length and complexity of their used prompts. We use the number of words to measure the prompt length and leverage DeepSeek-V3.1(Liu et al., 2024a) to score complexity of prompts from each benchmark. Details about the prompt template for complexity scoring could be seen in Appendix C.1.

| Benchmarks | Samples | Avg. Prompt Length | Complexity | | | |
| --- | --- | --- | --- | --- | --- | --- |
| | | | Semantic | Structure | Control | Avg. |
| EvalCrafter (Liu et al., 2024b) | 700 | 12.33 | 3.88 | 3.05 | 4.27 | 3.73 |
| VBench-Long (Huang et al., 2024b) | 946 | 7.64 | 2.75 | 2.11 | 2.76 | 2.54 |
| VBench 2.0-Complex Plot (Zheng et al., 2025) | 60 | 117.15 | 8.80 | 8.30 | 6.95 | 8.02 |
| VBench 2.0-Complex Landscape (Zheng et al., 2025) | 30 | 142.10 | 7.73 | _8.80_ | **8.50** | _8.34_ |
| VMBench (Ling et al., 2025) | 1050 | 26.23 | 5.96 | 5.36 | 4.39 | 5.24 |
| FilMaster-Complex (Huang et al., 2025) | 10 | 95.70 | _9.00_ | 8.00 | 7.20 | 8.07 |
| **LoCoT2V-Bench** *(ours)* | 240 | **236.66** | **9.01** | **8.98** | _8.25_ | **8.75** |

# 1 INTRODUCTION

In recent years, the rapid advancement of AI-Generated Content (AIGC) and the popularity of short-form video platforms have accelerated research on text-to-video generation. Current mainstream video generation models are able to produce short clips with high quality (OpenAI, 2024; Runway AI, 2025; DeepMind, 2025; MiniMax, 2025; Kuaishou, 2025; Kong et al., 2024; Wan et al., 2025). However, they struggle to generate long-form and complex videos, which we define in this work as videos longer than 30 seconds and typically under 60 seconds. To overcome this limitation, some works optimize model architectures and training strategies for longer sequences (He et al., 2022; Lu et al., 2024; Henschel et al., 2025; Chen et al., 2025a), while others leverage Large Language Models (LLMs) to plan scripts and orchestrate multiple tools for multi-shot or story-level video creation (Long et al., 2024; Zhuang et al., 2024; Xie et al., 2024; Zheng et al., 2024). These efforts have pushed the frontier of long video generation (LVG) forward.

Although recent advances have enabled more flexible approaches to long video generation, evaluating their performance under complex text inputs remains an open challenge. Researchers have made some progress in benchmarking video generation models (Huang et al., 2024a; Liu et al., 2024b; Ling et al., 2025; Han et al., 2025; Zheng et al., 2025). These work have proposed comprehensive evaluation framework via delicate prompt construction methodology and well-designed multi-dimensional metrics. However, most of them primarily target the evaluation of short videos. Their dependence on specific prompt construction strategies further limits their applicability to complex input scenarios, particularly when assessing longer videos in real-world settings. Moreover, existing benchmarks mainly emphasize visual quality, temporal consistency and prompt adherence, while overlooking higher-level aspects such as thematic expression and event-level coherence. This limitation becomes even more pronounced when evaluating long-form videos with richer content.

To address these gaps, we introduce LoCoT2V-Bench, a benchmark specifically designed for evaluating text-to-video generation under complex prompts and extended durations. **The key contributions of this work are as follows:**

- We construct a challenging prompt suite derived from diverse real-world videos, curated into 240 samples across 18 themes. Using powerful Vision-Language Models (VLMs) (Madaan et al., 2023), we generate longer and more complex prompts than existing benchmarks as shown in Table 1, explicitly incorporating scene transitions, camera motion, and event dynamics.
- We design a multi-dimensional evaluation framework that extends beyond conventional metrics such as visual fidelity, temporal consistency, and prompt adherence. Our framework introduces higher-level dimensions like thematic expression and novel event-level adherence, enabling fine-grained assessment of long-form video generation.
- We evaluate nine open-source LVG methods on five major dimensions composed of 26 sub-dimensions. Results show that while existing models excel in visual quality and overall consistency, they struggle with inter-event coherence, fine-grained prompt adherence, and narrative flow, etc. These findings provide actionable insights for future model development.

In summary, LoCoT2V-Bench provides the first systematic benchmark tailored for complex and long-form text-to-video generation. By combining realistic prompts, multi-dimensional metrics, and thorough empirical evaluation, it establishes a robust foundation for advancing LVG research.

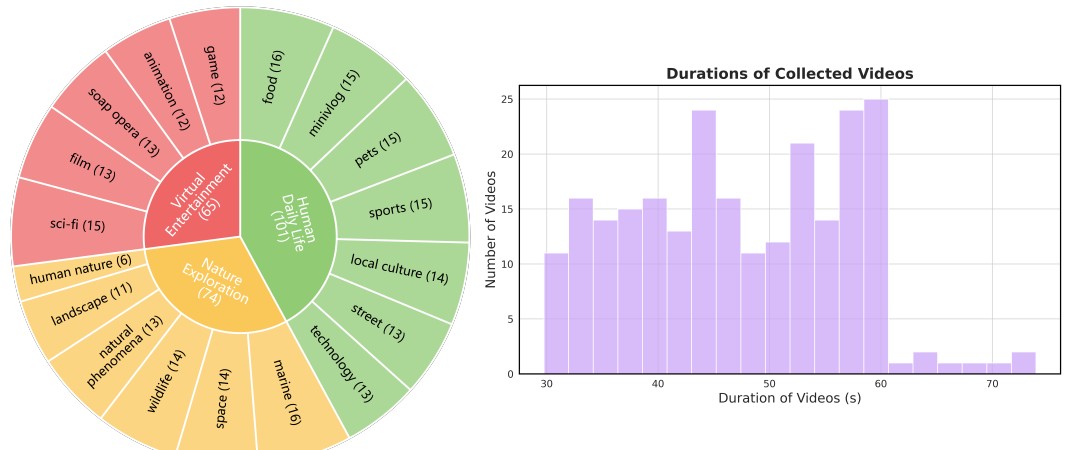

Figure 2: **Statistics of Collected Videos.** The two images demonstrate some statistics of our collected videos. *left:* Distribution of video quantity under different themes. *right:* Duration distribution of collected videos.

## 2 RELATED WORKS

**Long Video Generation** Long video generation requires models to generate videos longer than 10 seconds and it has always been an essential field in video generation area. Some of existing methods are mainly based on diffusion models (He et al., 2022; Wang et al., 2023; Lu et al., 2024; Ouyang et al., 2024; Song et al., 2025; Henschel et al., 2025). These studies introduce carefully designed modules and training strategies for extending short video generation models to longer video generation. Other methods use autoregressive models for LVG (Ge et al., 2022; Villegas et al.; Chen et al., 2024; Weng et al., 2024; Yin et al., 2025; Chen et al., 2025a; Teng et al., 2025). Due to autoregressive generation paradigm they could support variable length and even ultra-long video generation. While these investigations are capable of generating long videos with high quality, most of them are limited to single-scene video generation, narrowing their application scope.

Another type of long video, multi-scene video, has also achieved significant advance in recent years (Lin et al., 2023; Zhu et al., 2023; Long et al., 2024; Zhuang et al., 2024; Zheng et al., 2024). The considerable potential of LLM-driven agents towards tackling complex real-world problem has also brought some new thought into this area. For instance, (Xie et al., 2024; Wu et al., 2025b) utilize powerful multi-agent collaboration to simulate film production procedure in reality, enabling more attractive and complex multi-scene video generation.

**Video Generation Evaluation.** The great progress of video generation triggers the development of benchmarks for evaluating these methods. Traditional evaluation approaches focus on frame-level image quality and diversity, such as FID (Heusel et al., 2017), FVD (Unterthiner et al., 2019) and Inception Score (IS) (Salimans et al., 2016). CLIP-Score (Hessel et al., 2021) is also used to evaluate prompt adherence of generated videos. However, given that video evaluation inherently involves multiple factors, these metrics remain limited in scope and lack more comprehensive assessment.

To fully evaluate the quality of the generated videos, a series of benchmarks have been proposed recently (Huang et al., 2024a; Liu et al., 2024b; Ling et al., 2025; Qi et al., 2025; Yang et al., 2025). These works design prompt suites and leverage various tools to construct multi-dimensional evaluation metrics. While such efforts have led to comprehensive evaluation frameworks for video generation, they typically employ prompts describing a single scene with limited content and mainly target short video evaluation. For long-form and complex text-to-video generation, (Huang et al., 2024b) extends VBench (Huang et al., 2024a) to support longer videos, and (Zheng et al., 2025) further complements the evaluation suite with complex plot and landscape generation. However, the former still relies on simplified prompts, while the latter is constrained by the limited amount and complexity of its prompts. (Bugliarello et al., 2023; Zhuang et al., 2025) assesses videos from multi-prompt or multi-shot inputs primarily focus on story visualization rather than video generation.

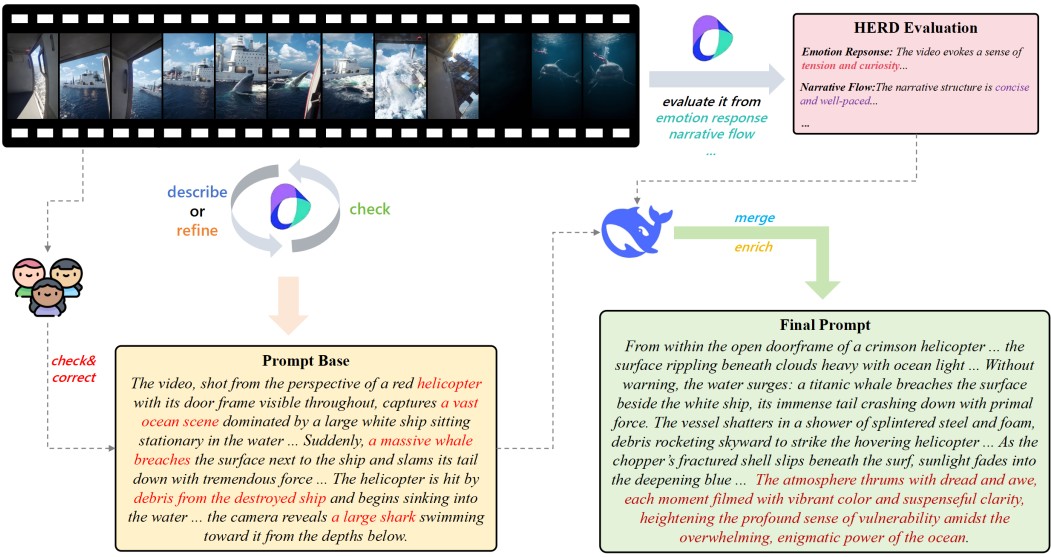

Figure 3: **A demonstration of the example-based workflow for prompt generation and evaluation integration:** We first use a powerful MLLM with self-refine (Madaan et al., 2023) to draft the prompt, which are then checked and corrected by experienced human verifiers based on the video source. HERD evaluation results are extracted from ground-truth videos by an MLLM. Finally we employ a strong LLM to merge them and enrich the integrated prompt result.

## 3 LoCoT2V-Bench

### 3.1 Prompt Suite Construction

As shown in Fig. 1, our prompt suite is constructed through a three-stage process comprising video collection, prompt generation, and evaluation information integration. We also provide an example for the last two stages to demonstrate the actual process as presented in Fig. 3.

**Video Collection.** To align our evaluation more closely with real-world video production, we collected thousands of short-form videos (30–60 seconds) from YouTube using yt-dlp[1] , guided by 18 predefined thematic keywords. We then manually filtered out invalid samples affected by subtitle or watermark occlusion, degraded visual quality, or misalignment between content and theme. As illustrated in Fig. 2, the final dataset consists of 240 videos, evenly distributed across 18 themes.

**Prompt Generation.** Given that more and more MLLM have demonstrated strong video understanding capacities (Bai et al., 2025; Zhang et al., 2025; Zhu et al., 2025; Guo et al., 2025), we employ them directly to generate raw prompts from the collected videos, rather than relying on human annotation. To ensure the quality of these prompts, we carefully design the generation instructions and adopt the self-refine (Madaan et al., 2023) paradigm for iterative optimization. In addition, we manually review each generated prompt and correct factual inaccuracies when necessary.

**Evaluation Information Integration.** To enable assessment along high-level aspects of video quality, we first introduce seven kinds of requirements: emotional response, narrative flow, character development, visual style, themes, interpretive depth and overall impression. These aspects will naturally compose one of our evaluation dimensions presented in the following section 3.2.5. And detailed descriptions of these dimensions are provided in Appendix B.8. Then we employ Seed1.5-VL (Guo et al., 2025) to evaluate each collected video across these dimensions. The resulting evaluations would be subsequently integrated into our previously generated raw prompts to obtain the final test prompts. As shown in Fig. 5 in the Appendix B.2 our prompts cover multiple elements such as light, camera and some spatial relationships (e.g. beneath) and mostly range from 200-300 words in length, ensuring their content richness.

---

[1]https://github.com/yt-dlp/yt-dlp

## 3.2 Evaluation Dimension Suite Construction

As shown in Fig. 1, we divide our evaluation dimensions into five categories partially inspired by some of the existing benchmarks (Huang et al., 2024a; Liu et al., 2024b): static quality, text-video alignment, temporal quality, content clarity and Human Expectation Realization Degree (HERD).

### 3.2.1 Static Quality (SQ)

Static quality focuses on frame-level image quality which is often separated into two parts: aesthetic quality and technical quality.

**Aesthetic Quality (AQ).** Aesthetic quality refers to the artistic and beauty value perceived by humans. We evaluate aesthetic quality of a generated video using the Aesthetic Predictor V2.5[2], a SigLIP-based (Zhai et al., 2023) predictor that assesses the aesthetics of an image on a scale from 1 to 10. Since videos consist of continuous frames, we sample one frame per second rather than scoring all frames, which substantially enhances evaluation efficiency. For normalization we take a more reasonable upper bound through the methods described in Appendix B.5.

**Technical Quality (TQ).** Technical quality refers to visual imperfections such as noise, blur, overexposure, and other artifacts that may degrade the viewing experience. We evaluate it using DOVER++ (Wu et al., 2023), a widely used model for user-generated video quality assessment (UGC-VQA). Given that DOVER was trained primarily on short videos with an average duration of approximately 10s, we segment each input long video into clips shorter than 10s and compute their average score as the technical quality score of the sample, ensuring more reliable results.

### 3.2.2 Text-Video Alignment (TVA)

Text-video alignment measures the adherence and faithfulness of the video content to the input text prompt. Considering the complexity of our prompts, we evaluate this dimension in a coarse-to-fine manner. Therefore, we divide text-video alignment into overall alignment and event-level alignment.

**Overall Alignment (OA).** Overall alignment assesses the global consistency between video and text. Existing benchmarks mostly rely on CLIP-based methods (Huang et al., 2024a; Liu et al., 2024b; Qi et al., 2025; Wu et al., 2025a). However, these methods are limited by CLIP's original design for learning image-text alignment features (Radford et al., 2021), which restricts their ability to handle more complex video-text evaluation. To address this, we leverage recent MLLMs with strong video understanding capability, together with embedding models that excel at encoding rich and complex textual semantics. Specifically we employ Qwen2.5-VL-7B (Bai et al., 2025) to generate a detailed description of the video and then compute the semantic similarity between this description and the prompt content of the video (See our used prompt for description in Appendix C.2). Note that we use the prompt base defined in Section 3.1, rather than the raw input prompt, to avoid interference from evaluation-related content. The resulting semantic similarity score serves as our final score.

**Event-level Alignment (EA).** Event-level alignment targets at assessing more fine-grained consistency between generated video and prompt text. We define an event as a combination of event description, subject, setting, action, and camera motion. As described in Section 3.1, we utilize DeepSeek-V3.1 (Liu et al., 2024a) to extract ground-truth events from the prompt base (See extraction prompt in Appendix C.3). Similarly, we extract event-level information from the detailed video description obtained in overall alignment and compute event-level similarity. Concretely, we first match generated and ground-truth events based on the semantic similarity of their event descriptions. Since the numbers and the relative positions of events that extracted from generated videos may vary significantly, we formulate this as a maximum-weight bipartite matching problem and solve it using the Hungarian algorithm (Kuhn, 1955). For each matched pair, we calculate both the overall semantic similarity of the event descriptions and the average field-level similarity across subject, setting, action, and camera motion. Their product serves as the event score.

To further incorporate temporal coherence, we penalize the score according to the disorder in event order. Specifically, let $I$ denote the number of inversions in the matched sequence and $I_{\max}$ the

---

[2]https://github.com/discus0434/aesthetic-predictor-v2-5

maximum possible inversions. The final event-level alignment score is defined as

$$S_{\text{event-align}} = \left(1 - \tfrac{I}{I_{\max}}\right) \cdot \frac{1}{N} \sum_{i=1}^{N} \left(S_{\text{semantic}}^{(i)} \times \tfrac{1}{4} \sum_{f \in \mathcal{F}} S_f^{(i)}\right), \ \mathcal{F} = \{subj, set, act, cam\}, \quad (1)$$

where $N$ is the number of matched event pairs. This formulation ensures that the score reflects both the fidelity of individual events and the correctness of their temporal order.

### 3.2.3 TEMPORAL QUALITY (TQ)

Temporal quality refers to the perceptual aspects of video in the time domain, emphasizing motion smoothness, frame-to-frame consistency, and the absence of temporal artifacts such as flicker, ghosting, or judder. In our work, we consider multiple aspects of temporal quality, including dynamic degree, motion smoothness, human action, and transition smoothness.

**Dynamic Degree & Motion Smoothness.** These two metrics were originally introduced in VBench (Huang et al., 2024a). The former judges whether the video contains significant motions to evaluate the degree of dynamics, while the latter assesses whether the motion in the generated video is smooth and consistent with physical laws in the real world. Given their universality and effect as demonstrated in the original paper, we directly adopt them for our evaluation.

**Human Action.** This metric aims to detect whether the preset human actions occur accurately and naturally in the video. We first employ LLMs with strong reasoning ability to identify whether human actions are present in each prompt and to extract detailed information such as subjects and action descriptions. Different from previous works that rely on action recognition models to detect specific action types (Huang et al., 2024a; Liu et al., 2024b), we adopt a multi-modal large language model (MLLM) with strong video understanding capability to evaluate both the occurrence and the smoothness of all actions (All used prompts are provided in Appendix C.4). This approach enables the evaluation of more complex actions and better aligns with our prompt settings.

**Temporal Flickering.** Temporal flickering refers to the instability of local regions or high-frequency details across consecutive frames, which leads to a perceivable flicker effect. For this dimension, we perform evaluation following the protocol of VBench (Huang et al., 2024a), which compute the mean absolute difference across static frames as final scores.

**Transition Smoothness.** Transition smoothness measures how gradual a scene change is. It is particularly important for multi-scene videos, yet has been rarely explored in prior work. To compute it, we first locate transition points using PySceneDetect[3] . Around each transition, we extract a temporal window and evaluate the similarity of frames based on pixel-level, structural, feature, and motion cues. The abruptness of each transition is then quantified as the variance of this similarity sequence, and the final video-level score is obtained by averaging over all detected transitions. The full formulation and implementation details are provided in Appendix B.6.

**Warping Error & Semantic Consistency.** These two metrics focus on temporal consistency of videos. Warping error measures pixel-level inconsistencies between consecutive frames after optical flow alignment, revealing temporal artifacts such as flickering or misalignment, while semantic consistency captures the stability of high-level semantics across frames, reflecting whether objects and scene meanings remain consistent over time. We employ the same evaluation methods as proposed in EvalCrafter (Liu et al., 2024b) for more reliable results.

**Intra- and Inter-event Temporal Consistency.** For assessing temporal consistency, we only consider Subject Consistency (SC) and Background Consistency (BC). They are designed to evaluate whether visual elements remain stable throughout a video. To make the assessment more fine-grained, we leverage the *event* structure introduced in our alignment evaluation (Section 3.2.2). Concretely, we first extract event clips using Time-R1-7B (Wang et al., 2025) and obtain frame-level subject crops and background regions with Grounded-SAM-2[4]. **Intra-event consistency** measures the similarity of the same element (subject or background) across frames within a single event, reflecting stability during continuous actions. **Inter-event consistency** measures the similarity of the same element across different events, reflecting stability under temporal gaps and scene changes.

---

[3]https://github.com/Breakthrough/PySceneDetect
[4]https://github.com/IDEA-Research/Grounded-SAM-2

For each video, we average the scores across all subjects or background regions to obtain overall intra- and inter-event consistency. Further implementation details are provided in Appendix B.7.

### 3.2.4 CONTENT CLARITY (CC)

Video generation often suffers from semantic inconsistencies, abrupt transitions, and thematic incoherence, motivating the need for a structured evaluation of content clarity. Therefore, we propose content clarity to assesses the semantic coherence and narrative quality of generated videos, with an emphasis on how effectively they convey meaningful information to viewers.

We propose a multi-dimensional assessment framework grounded in MLLMs with advanced video understanding capabilities. This framework is composed of four complementary dimensions: **Theme Clarity (TC)** to judge whether a central message is explicitly conveyed, **Logical Structure (LS)** to test temporal and causal coherence of scene progression, **Information Completeness (ICP)** to measure adequacy of visual context for comprehension, and **Information Consistency (ICS)** to assess coherence and alignment across shots.

For each video, the MLLM is prompted multiple times with controlled randomness to simulate diverse judgments like mean-of-score (MOS). Our used evaluation prompt could be found in Appendix C.5. Then each evaluation produces a structured output containing a score (0–4) and concise textual rationale for each evaluation dimension. The raw scores are normalized by the maximum scale and averaged across repeated trials to mitigate variance introduced by randomness. Finally the content clarity score for a video is computed as follows:

$$S(v) = \frac{1}{D} \sum_{d=1}^{D} s_d(v), \quad s_d(v) = \frac{1}{R} \sum_{i=1}^{R} \frac{s_d^{(i)}(v)}{4}, \tag{2}$$

where $s_d^{(i)}(v)$ denotes the raw score assigned to video $v$ on dimension $d$ at trial $i$, $R$ is the number of evaluation rounds, and $D = 4$ is the number of dimensions.

### 3.2.5 HUMAN EXPECTATION REALIZATION DEGREE (HERD)

Current benchmarks for video generation struggle to evaluate high-level dimensions such as emotional response and thematic expression. To address this limitation, we propose the Human Expectation Realization Degree (HERD), a framework that quantifies how well generated videos meet human expectations across these dimensions. As described in Section 3.1, we have incorporated multi-dimensional assessment information into our test prompts. Based on this information, we then generate multiple binary, content-specific questions for HERD evaluation of each dimension. Details about generation of these questions are provided in Appendix B.8.

The polarity of binary question matters as mentioned in FingER (Chen et al., 2025b), so we annotate the polarity of each question. Specifically, a question is labeled as "positive" if answering "yes" contributes positively to the final score, and "negative" if answering "no" does so. For example, "*Did the video make you feel tense and claustrophobic?*" is positive, whereas "*Did the characters lack depth and have unclear relationships?*" is negative. After that, each question is posed to a MLLM in a few-shot VQA setting, where the model responds with "yes", "no", or "unclear". For scoring, only "yes" and "no" responses are considered. The HERD score is defined as the proportion of polarity-consistent responses (i.e., "yes" for positive questions or "no" for negative ones) over the total number of valid (yes/no) responses, while "unclear" being treated invalid. This polarity-aware design ensures that the resulting scores more reliably reflect alignment with human expectations.

## 4 RESULTS & ANALYSIS

### 4.1 PERFORMANCE ON LOCOT2V-BENCH

To comprehensively demonstrate the capabilities of current long video generation methods, we select nine representative open-source methods as our baselines and perform evaluation on their generated videos. Detailed introduction and implementations about these methods are in Appendix B.1.

As shown in Table 2 and Fig. 8 in the Appendix B.10, existing LVG methods exhibit great performance in frame-level static quality. Nevertheless, they remain limited in the other four dimensions

Table 2: Performance results of all baseline methods evaluated over the five major dimensions. Scores for each dimension are obtained by averaging the scores of its corresponding sub-dimensions. Sub-dimension scores of temporal quality and HERD are presented in Table 3 and 4 perspectively. Note that all values are expressed as percentages to improve readability and conserve space.

| Methods | Static Quality | | | Text-Video Alignment | | | Temporal Quality | Content Clarity | | | | | HERD | Avg. |
|---|---|---|---|---|---|---|---|---|---|---|---|---|---|---|
| | AQ | TQ | Avg. | OA | EA | Avg. | | TC | LS | ICP | ICS | Avg. | | |
| FreeNoise (Qiu et al., 2023) | 65.38 | 71.34 | 68.36 | 63.28 | 47.17 | 55.23 | 73.26 | 71.32 | 72.36 | 71.53 | 80.38 | 73.90 | 50.00 | 64.15 |
| MEVG (Oh et al., 2024) | 41.75 | 18.33 | 30.04 | 64.66 | 49.13 | 56.90 | 66.70 | 45.38 | 45.59 | 46.67 | 55.45 | 48.27 | 47.54 | 49.89 |
| FreeLong (Lu et al., 2024) | 58.31 | 55.99 | 57.15 | 69.10 | 52.97 | 61.04 | 66.57 | 56.67 | 59.17 | 59.38 | 67.29 | 60.63 | 57.65 | 60.61 |
| FIFO-Diffusion (Kim et al., 2024) | 65.72 | 62.09 | 63.91 | 59.32 | 45.27 | 52.30 | 75.58 | 79.03 | 79.97 | 78.02 | 86.25 | 80.82 | 49.76 | 64.47 |
| DiTCtrl (Cai et al., 2025) | 54.27 | 62.13 | 58.20 | 71.70 | 54.42 | 63.06 | 70.77 | 59.90 | 63.09 | 60.87 | 69.27 | 63.28 | 60.72 | 63.21 |
| CausVid (Yin et al., 2025) | 62.44 | 89.54 | 75.99 | 73.30 | 59.64 | 66.47 | 69.84 | 57.43 | 60.14 | 61.42 | 68.33 | 61.83 | 63.55 | 67.54 |
| SkyReels-V2 (Chen et al., 2025a) | 67.20 | 80.18 | 73.69 | 70.98 | 46.33 | 58.66 | 79.49 | 71.39 | 72.71 | 71.18 | 79.20 | 73.62 | 62.74 | 69.64 |
| Vlogger (Zhuang et al., 2024) | 49.16 | 78.91 | 64.04 | 65.78 | 23.68 | 44.73 | 66.07 | 45.17 | 46.22 | 49.17 | 55.42 | 49.00 | 58.59 | 56.48 |
| VGoT (Zheng et al., 2024) | 85.50 | 96.79 | 91.15 | 67.07 | 42.83 | 54.95 | 71.21 | 78.92 | 78.13 | 77.78 | 84.31 | 79.79 | 63.74 | 72.17 |
| Ground-Truth | 61.93 | 59.58 | 60.76 | 78.79 | 57.50 | 68.15 | 71.90 | 64.38 | 67.15 | 65.17 | 72.19 | 67.22 | 66.92 | 66.99 |

Table 3: Performance results of all baseline methods on sub-dimensions of temporal quality. Note that all values are expressed as percentages to improve readability and conserve space. **Abbreviation Explanations:** "ITAE" refers to "Intra-event", "ITRE" refers to "Inter-event", "SC" refers to "Subject Consistency" and "BC" refers to "Background Consistency".

| Method | Dynamic Degree | Motion Smoothness | Warping Error | Semantic Consistency | Temporal Flickering | Transition Smoothness | Human Action | Event-level Consistency | | | | Avg. |
|---|---|---|---|---|---|---|---|---|---|---|---|---|
| | | | | | | | | ITAE SC | ITAE BC | ITRE SC | ITRE BC | |
| FreeNoise (Qiu et al., 2023) | 27.64 | 96.84 | 91.41 | 98.13 | 95.65 | 78.90 | 28.63 | 95.60 | 97.82 | 43.15 | 52.09 | 73.26 |
| MEVG (Oh et al., 2024) | 8.72 | 99.35 | 99.33 | 98.96 | 99.20 | 25.71 | 23.30 | 97.88 | 98.95 | 35.50 | 46.80 | 66.70 |
| FreeLong (Lu et al., 2024) | 23.30 | 98.41 | 94.85 | 98.50 | 98.07 | 18.24 | 31.59 | 91.16 | 98.34 | 37.41 | 42.40 | 66.57 |
| FIFO-Diffusion (Kim et al., 2024) | 62.21 | 96.29 | 88.92 | 97.44 | 94.23 | 73.49 | 23.61 | 94.37 | 97.22 | 45.93 | 57.72 | 75.58 |
| DiTCtrl (Cai et al., 2025) | 32.26 | 99.23 | 99.11 | 98.83 | 98.92 | 24.96 | 44.61 | 97.07 | 98.76 | 38.05 | 46.65 | 70.77 |
| CausVid (Yin et al., 2025) | 40.67 | 98.90 | 97.45 | 99.29 | 98.41 | 10.76 | 48.15 | 96.51 | 99.22 | 35.58 | 43.26 | 69.84 |
| SkyReels-V2 (Chen et al., 2025a) | 77.41 | 98.10 | 92.99 | 98.22 | 96.06 | 79.28 | 47.91 | 95.44 | 98.03 | 40.32 | 50.60 | 79.49 |
| Vlogger (Zhuang et al., 2024) | 36.70 | 95.35 | 81.54 | 96.12 | 94.27 | 27.03 | 37.46 | 90.31 | 96.48 | 34.28 | 37.20 | 66.07 |
| VGoT (Zheng et al., 2024) | 33.86 | 99.12 | 97.70 | 99.43 | 98.47 | 27.25 | 43.15 | 96.66 | 99.37 | 39.89 | 48.44 | 71.21 |
| Ground-Truth | 70.89 | 98.51 | 97.59 | 92.16 | 97.33 | 22.14 | 54.53 | 95.79 | 98.70 | 24.09 | 39.14 | 71.90 |

due to degradation in specific sub-dimensions. For example, their scores on event-level alignment are much lower than on overall alignment, indicating difficulty in capturing fine-grained semantic information. Apparently these methods could well preserve short-term temporal stability considering their excellent performance on some metrics like semantic consistency or intra-event consistency. However, they struggle to maintain long-term temporal consistency, as reflected by their low scores on inter-event consistency or transition smoothness. This may result from their weak capacities to model long-term context. In addition, their performance on high-level adherence remains unsatisfactory: they encounter substantial challenges in dimensions such as character development, narrative flow, and interpretive depth in HERD. These results suggest that current methods still lack the ability to generate complete and coherent long-form video content as expected by humans.

## 4.2 DOES VIDEO CONTENT TYPE IMPACT EVALUATION?

As shown in Fig. 2, we group the 18 prompt themes into three major categories to examine whether different methods exhibit significant performance variations across them. The results in Fig. 9 in the Appendix B.11 report the performance of these categories along the five evaluation dimensions introduced earlier. Overall, the differences across theme categories are minor, which aligns to some extent with the findings of EvalCrafter (Liu et al., 2024b), and further demonstrates the robustness of our evaluation framework to diverse content types. Nevertheless, several interesting patterns emerge. For example, methods achieve slightly higher text–video alignment scores on the nature exploration category. As shown in Fig. 6, prompts in this category are the shortest and of intermediate complexity among all themes, which may partly explain this observation.

## 4.3 ARE VIDEOS WITH HIGHER STATIC QUALITY PREFERRED?

A natural intuition is that higher static quality indicates better visual fidelity, which generally makes videos more visually appealing and easier to interpret. This advantage, however, may also influence the preference for certain video samples, thereby affecting the evaluation outcomes. To examine whether such bias occurred in our evaluation, we collected the scores of each test sample across the five dimensions and computed the linear correlations between static quality and each of the other four dimensions. The results are presented in Fig. 4 and Table 5.

Table 4: Performance results of all baseline methods evaluated on sub-dimensions of HERD. Note that all values are expressed as percentages to improve readability and conserve space.

| Method | Emotional Response | Narrative Flow | Character Development | Visual Style | Themes | Interpretive Depth | Overall Impression | Avg. |
|---|---|---|---|---|---|---|---|---|
| FreeNoise (Qiu et al., 2023) | 65.28 | 21.94 | 31.04 | 72.99 | 49.31 | 35.56 | 73.89 | 50.00 |
| MEVG (Oh et al., 2024) | 58.82 | 19.24 | 22.99 | 83.82 | 46.18 | 27.99 | 73.75 | 47.54 |
| FreeLong (Lu et al., 2024) | 74.31 | 25.83 | 33.06 | 84.03 | 64.37 | 37.92 | 84.03 | 57.65 |
| FIFO-Diffusion (Kim et al., 2024) | 68.68 | 24.58 | 28.75 | 73.89 | 45.62 | 34.10 | 72.71 | 49.76 |
| DiTCtrl (Cai et al., 2025) | 75.56 | 32.92 | 37.01 | 86.46 | 67.01 | 38.33 | 87.78 | 60.72 |
| CausVid (Yin et al., 2025) | 80.07 | 31.81 | 39.44 | 88.33 | 72.57 | **42.57** | 90.07 | 63.55 |
| SkyReels-V2 (Chen et al., 2025a) | 77.57 | 35.49 | 41.46 | 88.82 | 68.47 | 38.19 | 89.17 | 62.74 |
| Vlogger (Zhuang et al., 2024) | 71.32 | 28.54 | 34.58 | **90.35** | 68.26 | 29.37 | 87.71 | 58.59 |
| VGoT (Zheng et al., 2024) | **82.22** | 25.83 | **48.13** | 88.47 | 72.08 | 41.11 | 88.33 | 63.74 |
| Ground-Truth | 79.86 | **42.50** | 47.15 | 89.17 | **74.37** | 42.01 | **93.40** | **66.92** |

From this figure, we can see that static quality shows no strong linear correlation with the other four dimensions, suggesting that frame-level image fidelity exerts only a limited impact on metrics unrelated to visual quality. Notably, while text–video alignment is almost independent of static quality, content clarity exhibits the highest degree of correlation, which partly confirms our initial intuition. In addition, some sub-dimensions of temporal quality and HERD are inherently related to static quality, and their observed correlations are therefore reasonable. Overall, these results imply that potential biases introduced by static quality are controlled and limited, ensuring that the evaluation outcomes remain robust and representative.

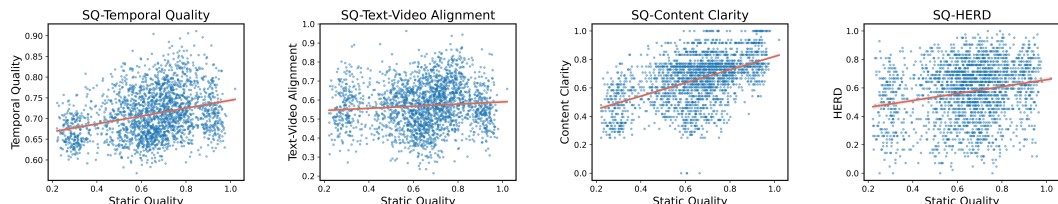

Figure 4: Visualization of correlation between static quality and other four dimensions. We display the results as four scatter plots and their regression lines. Note that "SQ" here refers to Static Quality.

## 4.4 ENTANGLEMENT BETWEEN EVENT-LEVEL ALIGNMENT AND TEMPORAL CONSISTENCY

As discussed in Section 3.2.2 and Section 3.2.3, we evaluate both event-level prompt adherence and temporal consistency for fine-grained assessment. This requires extracting clips for each event or retrieving images of individual subjects from generated videos. However, when the generated content fails to satisfy the input prompt, the corresponding subjects or events may not appear at all, which inevitably impact the measured temporal consistency. Such cases suggest a potential entanglement between event-level alignment and temporal consistency. Therefore, we further investigate whether this entanglement is present in our results.

As shown in Table 5, event-level alignment scores of all test samples exhibit low linear correlations with event-level temporal consistency, particularly when compared with inter-event subject and background consistency. This outcome can be explained by the different characteristics of the models used in these evaluation dimensions. Even when a VLM determines that certain generated videos are poorly aligned with their prompts, the TVG models and the semantic segmentation models may still successfully extract clips related to the event description or images corresponding to the described subject. These extracted images can remain highly coherent across consecutive frames of retrieved clips, leading to relatively low alignment scores but high consistency scores.

## 4.5 HOW DOES COMPLEXITY OF PROMPTS INFLUENCE EVALUATION?

Given that our evaluation scenario involves input prompts that are generally longer and more complex than typical cases, we further investigate how evaluation outcomes vary with different levels of prompt complexity. Specifically, we analyze the relationships between three types of complexity (semantic, structural, and control) and the five evaluation dimensions across all test samples generated by nine baseline methods. The results are shown in Fig. 10 in the Appendix B.12.

Table 5: Correlation results across evaluation metrics. Three representative methods are used to assess linear correlations between pairs of metrics based on sample-level score sequences.

| Metric 1 | Metric 2 | Pearson's $r$ | Spearman's $\rho$ | Kendall's $\tau$ |
|---|---|---|---|---|
| Static Quality | Temporal Quality | 0.2942 | 0.2924 | 0.1937 |
| | Text-Video Alignment | 0.0841 | 0.1093 | 0.0701 |
| | Content Clarity | 0.4977 | 0.4853 | 0.3401 |
| | HERD | 0.2136 | 0.1958 | 0.1328 |
| Event-level Alignment | Intra-event Subject Consistency | 0.1933 | 0.2381 | 0.1619 |
| | Intra-event Background Consistency | 0.3386 | 0.2769 | 0.1901 |
| | Inter-event Subject Consistency | 0.0121 | 0.0097 | 0.0048 |
| | Inter-event Background Consistency | 0.0110 | -0.0008 | 0.0011 |

We observe that prompt complexity indeed affects evaluation outcomes. In particular, methods tend to perform worse with higher semantic complexity, indicating their difficulty in understanding complex semantics. Similar trends are found with structural complexity. By contrast, the effect of control complexity is less pronounced. We attribute this to the fact that baseline methods are generally less sensitive to stylistic, technical, dynamic, and consistency-related control elements, and often fail to meet such requirements. Moreover, we find that certain dimensions—such as static quality and text–video alignment—are less influenced by prompt complexity, suggesting that they may better reflect the inherent capability of generation methods.

## 5 CONCLUSION

We introduced LoCoT2V-Bench, a benchmark specifically designed for long-form and complex text-to-video generation. By constructing prompts from real-world videos and developing a multi-dimensional evaluation suite composed of five representative dimensions, our framework enables fine-grained and holistic assessment beyond existing benchmarks. Extensive experiments on nine representative methods reveal that while current approaches perform well in visual fidelity and short-term temporal stability, they struggle with fine-grained event alignment, long-range temporal coherence, and high-level narrative adherence. Analyses of content types, prompt complexity, and metric entanglement further highlight both the robustness of LoCoT2V-Bench and the challenges faced by current models. We envision our benchmark as a foundation for rigorous evaluation and as guidance for future research toward generating long-form videos that are not only visually compelling but also coherent, controllable, and aligned with human expectations.

## 6 ETHICAL STATEMENT

All video data in LoCoT2V-Bench are collected from YouTube using yt-dlp in compliance with the platform's terms of service and copyright regulations. A rigorous filtering process, combining automatic checks and manual review, was applied to exclude invalid or harmful content. Prompts were generated and refined using VLMs and LLMs under strict instructions prohibiting PII, offensive, violent, or otherwise inappropriate material, with additional human verification to ensure factual accuracy and ethical compliance. During our evaluation, no private or sensitive data were used and all procedures adhere to relevant ethical guidelines for AIGC research, ensuring LoCoT2V-Bench promotes safe and responsible development of long-form text-to-video generation technology.

## 7 REPRODUCIBILITY STATEMENT

We will release the prompt data, evaluation code, and benchmark results of LoCoT2V-Bench at [https://anonymous.4open.science/r/LoCoT2V-Bench-1518/]. An initial draft of the constructed prompt suite in JSON format has already been provided. Due to the current complexity of the project structure, the full release will be made after code reorganization and thorough verification to ensure correctness and usability in a timely manner.

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

## A  APPENDIX

**Main Content of the Appendix:**

1. B Supplementary Details about LoCoT2V-Bench Implementations
2. C Prompt Template Used in Some Evaluation Methods
3. D Numerical Results
4. E Case Study
5. F Detailed LLM Usage in Our Work

## B  SUPPLEMENTARY DETAILS ABOUT LoCoT2V-BENCH IMPLEMENTATIONS

### B.1  BASELINE METHODS INTRODUCTION

For a comprehensive evaluation of current long video generation techniques, we select nine representative open-source methods based on their availability, popularity in the community, and diversity in modeling strategies. The selected methods are listed as follows. Given that some of them rely on multi-prompt input such as MEVG and FreeNoise, we leverage DeepSeek-V3.1(Liu et al., 2024a) to generate event-level prompt based on original prompt and extracted events mentioned in Section 3.2.2. These event-level prompts are then used for multi-prompt input.

- **FreeNoise** (Qiu et al., 2023) proposes a tuning-free paradigm for longer video generation with pretrained diffusion models by rescheduling initial noise to maintain long-range temporal coherence, plus a motion-injection trick to support multi-prompt conditioning, achieving superior quality.

- **MEVG** (Oh et al., 2024) is a training-free pipeline that turns a pre-trained T2V diffusion model into a multi-prompt storyteller. It uses an LLM prompt generator to split a long story into single-event captions and injects dynamic noise and last-frame inversion to initialize each new clip from the previous last frame, then applies structure-guided sampling to keep frames within a clip coherent.

- **FreeLong** (Lu et al., 2024) proposes a training-free SpectralBlend Temporal Attention mechanism: it fuses the low-frequency parts of global features (for overall coherence) with the high-frequency parts of local features (for fine detail) via 3-D FFT, enabling a 16-frame diffusion model to generate 128-frame videos with better consistency and fidelity.

- **FIFO-Diffusion** (Kim et al., 2024) enables a pretrained short-clip diffusion model to generate endless videos without retraining by performing diagonal denoising in a small FIFO frame queue, where noise increases toward the tail while the clean head is popped and new noise is pushed. To bridge the gap with uniform-noise training and reduce memory usage, it further introduces latent partitioning and lookahead denoising, achieving high-quality, temporally coherent long video generation.

- **DiTCtrl** (Cai et al., 2025) proposes a tuning-free approach for long video generation from multiple text prompts based on the MM-DiT architecture. By analyzing and leveraging its attention mechanism, it achieves smooth transitions and consistent motion via a novel KV-sharing strategy and latent blending.

- **CausVid** (Yin et al., 2025) is a fast autoregressive video diffusion model distilled from a slow bidirectional teacher using asymmetric distribution matching distillation (DMD), reducing generation latency from 219 s to 1.3 s and enabling streaming 9.4 FPS video on one GPU while maintaining state-of-the-art quality.

- **SkyReels-V2** (Chen et al., 2025a) synergizes an MLLM-based captioner, multi-stage pre-training, motion-specific reinforcement learning, and a diffusion-forcing framework to generate infinite-length, cinematic-quality videos while achieving state-of-the-art prompt adherence and motion fidelity. We use its 540P version in our practice.

- **Vlogger** (Zhuang et al., 2024) proposes an LLM-directed pipeline that decomposes a long vlog into four stages—Script, Actor, ShowMaker, Voicer—and introduces a new diffusion model (ShowMaker) that conditions on both text and actor images to generate coherent, variable-length scenes. It can produce 5-minute vlogs from open-world text without extra long-video training, setting a new zero-shot baseline for long video generation.

- **VGoT** (Zheng et al., 2024) is a training-free modular framework for multi-shot video generation. It decomposes the process into four collaborative modules: script generation, keyframe creation, shot-level video synthesis, and cross-shot smoothing. It ensures narrative coherence and visual consistency across shots using structured cinematic prompts and identity-preserving embeddings.

### B.2 PROMPT SUITE STATISTICS

Here we give some basic statistics about our constructed prompt suite. We illustrate the general content of the prompts in LoCoT2V-Bench through a word cloud figure and display the length distribution plot as well. The results can be seen in Fig. 5.

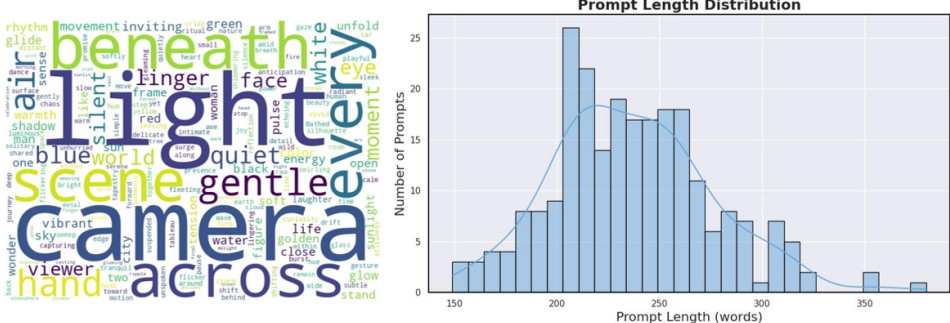

Figure 5: **Prompt Suite Statistics.** The two graphs demonstrate some statistics of our prompt suite. *left:* the word cloud to visualize word distribution of our prompts. *right:* the prompt length distribution of our prompt suite measured by the number of words.

### B.3 PROMPTS FROM DIFFERENT THEME CATEGORIES

As mentioned in Section 4.2, we assume that the variation of prompt complexity among different prompt content types may lead to the performance difference on assessment of text-video alignment. To verify this assumption, we display the average length and complexity of prompts in different themes. Results are shown in Fig. 6.

Figure 6: Length and complexity of prompts from different theme categories. We use 500 as the upper bound for average length while 10 for complexity and execute normalization based on them.

### B.4 MEASURING THE COMPLEXITY OF EACH PROMPT

As shown in Table. 1 we provide a complexity score composed of three dimensions. The complexity score of each prompt is directly obtained by DeepSeek-V3.1 (Liu et al., 2024a) with the following definitions for three dimensions. The prompt we used for scoring is provided in Appendix C.1.

- **Semantic Complexity** pertains to the semantic elements within a prompt, including entities and the relationships among them. This dimension necessitates that models accurately interpret the events and interactions specified in the prompt.

- **Structural Complexity** mainly focuses on the manner where prompts convey their content, facilitating diverse textual expressions and structured organization. Such complexity challenges models' capacity to process and adapt to flexible inputs.

- **Control Complexity** concentrates on constraints imposed on the outputs of generative models. Users may, for instance, specify requirements regarding visual style, camera motion, or the presence of specific objects. As such, this dimension is intended to capture these elements in prompts and assess whether models are able to fulfill these requirements.

### B.5 DERIVING THE UPPER BOUND OF THE AESTHETIC QUALITY SCORE

Although Aesthetic Predictor V2.5 can effectively assess the aesthetic quality of images and provide reasonable scores, its preset upper bound (10.0) is rarely attainable in practice. Even high-resolution images with strong visual appeal typically receive scores around 8, as shown in Fig. 7, which motivates us to establish a more appropriate reference upper bound. To this end, we introduce a Relative Reference Upper Bound (RR-UB), derived from high-quality image datasets rather than relying on an arbitrary fixed maximum.

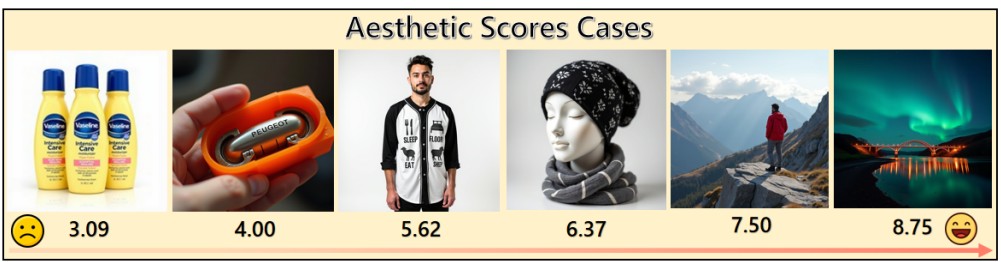

Figure 7: Examples of images corresponding to different aesthetic scores predicted by Aesthetic Predictor V2.5. The images are extracted from the data_1024_10K subset of the Text-to-Image-2M dataset. For clarity, scores are rounded to two decimal places in the figure.

Specifically, we use the Text-to-Image-2M[5] dataset to compute RR-UB. This dataset consists of approximately 2 million curated text–image pairs designed for fine-tuning text-to-image models. It includes two subsets: data_512_2M, composed of several high-quality image collections, and data_1024_10K, containing 10K images generated by Flux-dev[6] with GPT-4o (Hurst et al., 2024) prompts. We adopt the latter subset as the basis for RR-UB, assigning scores to each image and computing the mean of the top 10% as the final reference value. In this way, we can more reliably evaluate how closely the frame-level visuals of generated videos align with high-quality images, and provide a more appropriate upper bound than the default maximum score of 10.

### B.6 COMPUTING TRANSITION SMOOTHNESS

Here we give a more detailed explanation about how we compute transition smoothness score for a video. As we first obtain transition points $t_i$ via PySceneDetect, we then consider a temporal window of $2k + 1$ frames, from $t_i - k$ to $t_i + k$. For each frame $f_j$ in the window, we compute a similarity score $s_j$ as a weighted sum of four normalized features:

$$s_j = \alpha_1 \hat{s}_{mae} + \alpha_2 \hat{s}_{ssim} + \alpha_3 \hat{s}_{siglip} + \alpha_4 \hat{s}_{mc}, \quad \alpha_1 + \alpha_2 + \alpha_3 + \alpha_4 = 1, \tag{3}$$

where $\hat{\cdot}$ denotes feature-wise normalization within the window, $s_{mae}$ is pixel-level mean absolute error, $s_{ssim}$ is structural similarity, $s_{siglip}$ is feature similarity, and $s_{mc}$ is motion consistency estimated with RAFT (Teed & Deng, 2020). This gives a similarity sequence $\{s_j\}_{j=1}^{2k}$. We define the transition abruptness at $t_i$ as the normalized variance of the sequence:

$$A(t_i) = \frac{\text{Var}(S(t_i)) * b}{\text{Var}(S(t_i)) * b + c}, \quad S(t_i) = \{\hat{s}_j | j = 1, ..., 2k\}, \tag{4}$$

where $\hat{s}_j = s_j / \sum_{j=1}^{2k} \{s_j\}$, $b$ is a scaling factor, and $c$ is a small constant to stabilize the denominator. Finally, the transition smoothness is defined as:

$$S(t_i) = 1 - A(t_i). \tag{5}$$

Finally, to obtain a video-level score, we aggregate transition smoothness across all detected transitions. Given a set of transition points $\{t_i\}_{i=1}^{N_1}$, the overall smoothness is defined as the average of individual transition scores:

$$Score_{ts} = \frac{1}{N_1} \sum_{i=1}^{N_1} S(t_i). \tag{6}$$

### B.7 IMPLEMENTATION DETAILS ABOUT EVENT-LEVEL TEMPORAL CONSISTENCY

Here we demonstrate how we compute event-level temporal consistency scores through a mathematical description. Let the video be partitioned into $K$ events $\mathcal{E} = \{e_1, \ldots, e_K\}$, where each event $e$ contains a set of frames $F_e = \{f_{e,1}, \ldots, f_{e,T_e}\}$. For each frame $f_{e,t}$ we apply Grounding-SAM-2 to

---

[5]https://huggingface.co/datasets/jackyhate/text-to-image-2M
[6]https://huggingface.co/black-forest-labs/FLUX.1-dev

extract subject masks $\{M_{e,t}^s\}$, and obtain subject crops $C_{e,t}^s$ (masked regions of the original frame). We then compute unit-normalized visual feature vectors:

$$\phi_{e,t}^s = \text{norm}\left(\text{Encoder}(C_{e,t}^s)\right) \in \mathbb{R}^d, \qquad \|\phi_{e,t}^s\|_2 = 1, \tag{7}$$

**Intra-event Subject Consistency.** Suppose subject $s$ appears in event $e$ at frame indices

$$T_{e,s} = \{t_1 < t_2 < \cdots < t_n\}. \tag{8}$$

The intra-event consistency is defined as the average similarity between consecutive frames in the sequence. Let

$$A_{e,s} = \{(t_k, t_{k+1}) \mid k = 1, \ldots, n-1\}. \tag{9}$$

If $n \geq 2$, then

$$C_{e,s}^{\text{intra}} = \frac{1}{|A_{e,s}|} \sum_{(u,v) \in A_{e,s}} \text{sim}\left(\phi_{e,u}^s, \phi_{e,v}^s\right). \tag{10}$$

We use the cosine similarity function as the $sim(\cdot)$. The event-level score is the mean across all subjects in the event:

$$C_e^{\text{intra}} = \frac{1}{|\mathcal{S}_e|} \sum_{s \in \mathcal{S}_e} C_{e,s}^{\text{intra}}, \tag{11}$$

and the video-level score is obtained by averaging across events (with optional weighting by event length):

$$C_{\text{video}}^{\text{intra}} = \frac{1}{\sum_e w_e} \sum_e w_e \, C_e^{\text{intra}}. \tag{12}$$

**Inter-event Subject Consistency.** Let subject $s$ appear in $m$ different events $E_s = \{e_1, \ldots, e_m\}$, with frame sets $T_{e_i,s}$. We define the cross-event similarity between two events as

$$S_{e_i,e_j}^s = \frac{1}{|T_{e_i,s}| \, |T_{e_j,s}|} \sum_{t \in T_{e_i,s}} \sum_{u \in T_{e_j,s}} \text{sim}\left(\phi_{e_i,t}^s, \phi_{e_j,u}^s\right). \tag{13}$$

The inter-event consistency of subject $s$ is the average across all event pairs:

$$C_s^{\text{inter}} = \frac{2}{m(m-1)} \sum_{1 \leq i < j \leq m} S_{e_i,e_j}^s. \tag{14}$$

The video-level score averages across all subjects that appear in at least two events:

$$C_{\text{video}}^{\text{inter}} = \frac{1}{|\mathcal{S}'|} \sum_{s \in \mathcal{S}'} C_s^{\text{inter}}. \tag{15}$$

**Background Consistency.** Background consistency is computed analogously to the subject case. For each frame, subject regions are removed using the masks $\{M_{e,t}^s\}$ (optionally dilated to avoid residual edges), yielding

$$B_{e,t} = f_{e,t} \odot \left(1 - \bigcup_s M_{e,t}^s\right). \tag{16}$$

This approach is consistent with that adopted in SkyReels-A2 (Fei et al., 2025). We then extract normalized features $\psi_{e,t} = \text{norm}(\text{Encoder}(B_{e,t}))$ and compute intra- and inter-event background consistency by replacing $\phi$ with $\psi$ in the above definitions.

## B.8 HERD DIMENSION CONSTRUCTION AND EVALUATION

Here we explain the whole process of how we construct and evaluate proposed HERD dimension. We first give an introduction about all dimensions included in HERD metrics as follows:

- **Emotional Response** assesses the emotional impact of the video—whether it evokes curiosity, tension, inspiration, or confusion—and examines how effectively it engages viewers' feelings and maintains their emotional attention throughout.

- **Narrative Flow** examines the clarity and coherence of the storyline, including scene transitions and pacing, focusing on whether the narrative unfolds smoothly, feels rushed, or allows moments for reflection.

- **Character Development** evaluates the depth, authenticity, and consistency of characters, as well as the evolution of their relationships, emphasizing how these elements contribute to audience engagement and narrative believability.

- **Visual Style** analyzes the use of cinematography, color palette, lighting, and framing in establishing mood, atmosphere, and tone, considering how visual choices enhance story immersion and emotional resonance.

- **Themes** reflects on the underlying ideas, messages, or social commentary, assessing whether they are clearly expressed, thought-provoking, and meaningfully integrated with the video's overall narrative and intent.

- **Interpretive Depth** considers the degree of ambiguity, symbolism, and openness to multiple interpretations, evaluating whether the video encourages reflection, discussion, and a deeper engagement beyond the surface narrative.

- **Overall Impression** captures the lasting effect of the video, considering its overall impact, memorability, and appeal, and reflecting on its entertainment, educational, or emotional value for a broad range of audiences.

**HERD Questions Generation** To obtain target information for evaluating these dimensions with respect to each test prompt, we employ Qwen2.5-VL-72B (Bai et al., 2025) to generate dimension-wise assessment results based on the real-world videos collected in Section 3.1. Subsequently, DeepSeek-V3.1 (Liu et al., 2024a) is used to generate corresponding questions from these results. To mitigate randomness and format preference issues inherent to LLMs, we design multiple questions per dimension and, in practice, set six questions for each dimension of HERD.

**HERD Questions Polarity Annotation** While the generated outcomes are generally satisfactory, we observe that our initial scoring method—simply calculating the proportion of "yes" responses—introduces bias. Specifically, a positive answer may, in some cases, correspond to a negative contribution in the intended dimension. To correct this, we prompt DeepSeek-V3.1 to annotate the polarity of each question with respect to the evaluation requirements. During scoring, a "yes" response increases the score only when it aligns with the annotated polarity. This adjustment ensures that the HERD scores are more robust and reliable. Details about prompts that input into DeepSeek-V3 to get our expected results are provided in Appendix C.6.

### B.9 SUPPLEMENTARY EXPLANATION FOR EVALUATION METRICS FROM EXISTING WORKS

As a matter of fact we do adopt some metrics from existing video generation benchmarks. Therefore, we provide more detailed description for these metrics as follows to serve as complementary information of the corresponding parts in the main body:

- **Dynamic Degree** This metric, introduced in VBench (Huang et al., 2024a), evaluates whether a generated video contains observable motion. It utilizes RAFT (Teed & Deng, 2020) to estimate optical flow between consecutive frames and computes the mean of the top 5% flow magnitudes to classify videos as dynamic or static. The dynamic degree is then defined as the proportion of generated videos that are non-static.

- **Motion Smoothness** Originating from VBench (Huang et al., 2024a), this metric evaluates the temporal smoothness of generated video motion. It leverages the motion prior of video frame interpolation models, which assume short-term real-world motion to be approximately linear or quadratic. Given a frame sequence of a generated video $[f_1, f_2, \ldots, f_{2n}]$, all odd-indexed frames are removed to form a low-frame-rate sequence, and an interpolation model is used to reconstruct the missing frames $[\hat{f}_1, \hat{f}_2, \ldots, \hat{f}_{2n-1}]$. The mean absolute error (MAE) between reconstructed and original frames is then computed and normalized as

$$S_{MAE-norm} = \frac{255 - S_{MAE}}{255}. \tag{17}$$

The resulting score lies in $[0, 1]$ with higher values indicating smoother, more physically consistent motion.

- **Warping Error** This metric stems from EvalCrafter (Liu et al., 2024b) and measures the temporal consistency between consecutive frames. Following prior blind temporal consistency method (Lai et al., 2018; Lei et al., 2020; Qi et al., 2023), it first estimate the optical flow between each pair of adjacent frames using a pre-trained optical flow network (Teed & Deng, 2020). The earlier frame is then warped to the later frame according to the estimated flow. Temporal inconsistency is quantified as the pixel-wise difference between the warped frame and the actual subsequent frame. The final warping error is obtained by averaging these differences over all frame pairs, where lower values indicate better temporal consistency. However, we use its negative logarithmic value as the final result to make the scores positively correlated with model performance.

- **Semantic Consistency** Also proposed in EvalCrafter (Liu et al., 2024b), this metric focuses on semantic consistency between adjacent frames. It consider the cosine similarity of the semantic features of each two consecutive frames ($feat(f_t), feat(f_{t+1})$) and take the average as the final score. We utilize the SigLIP2 (Tschannen et al., 2025) instead of CLIP (Radford et al., 2021) in the original paper to obtain better features of each frame.

### B.10 VISUALIZATION FOR MAIN RESULTS

To better illustrate the performance gaps among different baseline methods across five major dimensions, we provide radar figures based on the numerical results in Table 2, 3 and 4.

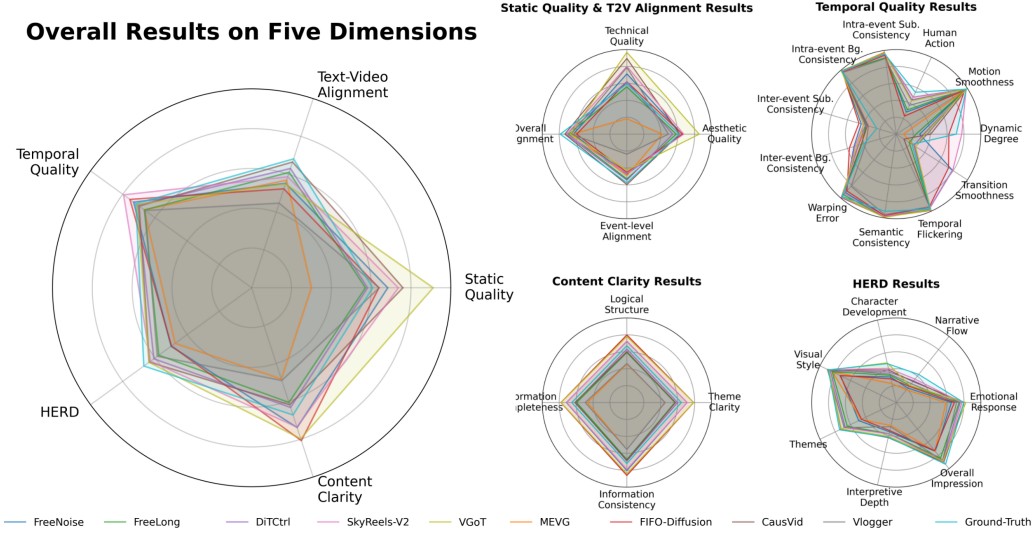

Figure 8: Evaluation results of all baselines on LoCoT2V-Bench. Results of five main dimensions and their sub-dimensions are presented together.

### B.11 VISUALIZATION RESULTS FROM DIFFERENT THEME CATEGORIES

In this section, we present evaluation results of the samples, which are grouped into three theme-based categories and assessed along five major dimensions. The results are illustrated using three parallel radar plots in Fig. 9, each corresponding to one theme category.

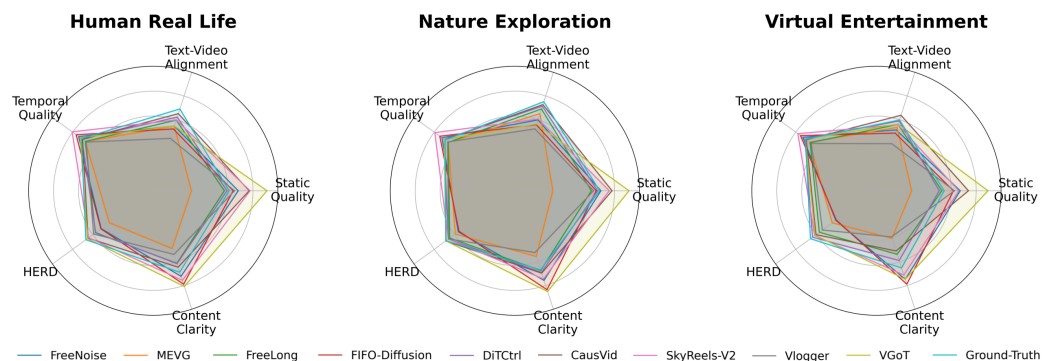

Figure 9: Evaluation results of samples from different prompt content theme categories.

### B.12 CORRELATIONS BETWEEN PROMPT COMPLEXITIES AND EVALUATION RESULTS

Here we provide our experiment results mentioned in Section 4.5. As shown in Fig. 10, we illustrate our results in the form of 15 violin plots and omit the scatters for better visualization effect. Each row corresponds to the relationship between a prompt complexity type and different evaluation dimensions, and each column corresponds to the relationship between a given evaluation dimension and different types of prompt complexity.

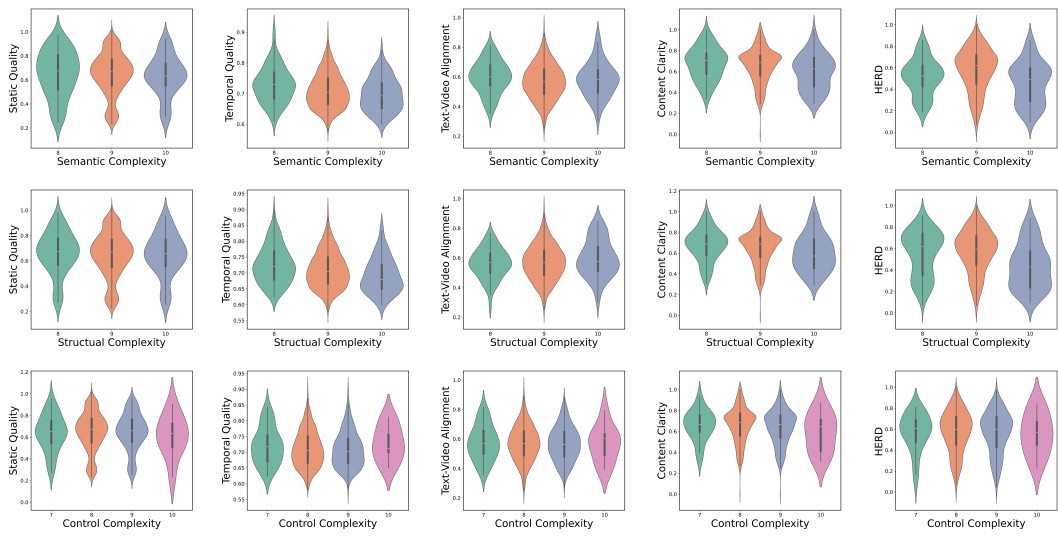

Figure 10: Correlation results between three types of prompt complexities mentioned in Table 1 and evaluation results on five dimensions. Scatters of samples are omitted for better visualization.

## C PROMPT TEMPLATE USED IN SOME EVALUATION METHODS

### C.1 COMPLEXITY SCORING PROMPT

**Complexity Scoring Prompt**

```
You are an expert evaluator of prompts used for image or video generation.
Your task is to analyze the complexity of a given prompt in detail.
Return the output strictly as a JSON object in the following nested dictionary
structure:
{
```

```
    "semantic_complexity": {
        "score": <integer 1-10>,
        "explanation": "<short explanation>"
    },
    "structural_complexity": {
        "score": <integer 1-10>,
        "explanation": "<short explanation>"
    },
    "control_complexity": {
        "score": <integer 1-10>,
        "explanation": "<short explanation>"
    }
}

### Evaluation criteria ###

1. Semantic complexity:
   - Number of entities (subjects, objects, characters).
   - Number of attributes or modifiers.
   - Abstract or metaphorical concepts.
   - Relationships or interactions between entities.

2. Structural complexity:
   - Prompt length and density.
   - Nested or hierarchical descriptions.
   - Logical relations (conditions, causality, comparisons).
   - Scene richness (multiple settings or sub-elements).

3. Control complexity:
   - Artistic or stylistic constraints (anime, cyberpunk, Van Gogh, etc.).
   - Technical constraints (camera angle, lens type, lighting).
   - Temporal dynamics (video actions, transitions).
   - Consistency requirements (identity or object continuity).
   - Explicit numeric or technical parameters.

### Few-shot examples ###

**Example 1 (simple prompt):**
[Example 1]

**Example 2 (moderately complex prompt):**
[Example 2]

**Example 3 (highly complex prompt):**
[Example 3]

### Now evaluate the following prompt:
{prompt_text}
```

## C.2    OVERALL DESCRIPTION GENERATION PROMPT

**Overall Description Generation Prompt**

```
## System ##
You are a highly capable visual understanding assistant, skilled in analyzing and
summarizing video content with precision and clarity.

## Task ##
Your goal is to produce a coherent and clear paragraph that accurately summarizes
the content of a given video.
```

```
Please follow these steps internally (do not output intermediate results):

1. **Event Detection**: Identify all major events in the video and arrange them in
chronological order.
2. **Visual Element Analysis**: For each event, identify and deeply analyze the key
visual components by specifying their attributes and visual characteristics:
   - Subjects: Identify each subject (e.g., person, animal, object) and describe
   their appearance (e.g., clothing, facial expression, posture, size, color,
   design).
   - Environments: Describe the setting in detail, including lighting conditions,
   spatial layout, textures, atmosphere, and any notable background elements.
   - Actions: Detail the actions with clarity-specify how movements are performed
   (e.g., slow vs. rapid, smooth vs. abrupt), gestures, and interaction between
   subjects or with objects.
   - Camera Dynamics: Describe how the camera behaves visually-note the type, speed,
   and purpose of camera movements (e.g., a slow pan to build suspense, a sudden
   zoom to highlight surprise), including angle perspectives and focal changes.
3. **Event Description**: Describe each event accurately, incorporating the visual
elements identified.
4. **Summary Composition**: Integrate all event descriptions into a single,
well-structured paragraph that captures the full sequence and essence of the video.

## Output Format ##
Only output the final summary paragraph. Do not include any intermediate steps,
bullet points, or reasoning process.
```

## C.3 EVENT EXTRACTION PROMPT

**Event Extraction Prompt**

```
You are an event extraction assistant. You will be given a textual description of a
video. Your task is to extract all the key events from the description in the order
they occur.

Each event should be represented as a JSON object with the following five fields:

- "event": A natural-language description of the event as a whole. Ensure the
descriptions are contextually coherent with each other and reflect a consistent
narrative flow across the video.
- "subject": Who or what is involved in the event. If there is no clear visual
subject (e.g., a landscape or object), use an empty string "" as default
- "setting": Where the event takes place
- "action": What happens during the event
- "camera motion": How the camera moves or is positioned (e.g., static, panning,
zoom-in, tracking). If not explicitly stated, infer it or use "static"

Make sure the events together form a continuous and coherent sequence, as if
telling a consistent visual story from beginning to end. Avoid treating them as
isolated incidents.

When describing each event, feel free to refer back to earlier subjects or settings
if the video appears to maintain continuity.

Please only output a **JSON array** of all events in the order they occur as
response, and do not include any irrelevant information:

```json
[
  {
```

```
    "event": "...",
    "subject": "...",
    "setting": "...",
    "action": "...",
    "camera motion": "..."
  },
  ...
]
```

Here is the video description:
{description_text}
```

## C.4   PROMPTS RELATED TO HUMAN ACTION EXTRACTION

**Human Action Extraction Prompt**

```
You are an assistant that analyzes video generation prompts.
Your task is to detect whether the prompt contains any *human action* (a human
subject performing a specific action).

## Output format ##

* Always output a list.
* Each element in the list is a JSON object with two fields:
  * `"subject"`: the human subject
  * `"action"`: the action performed
* If no human action is detected, return `[]`.

## Examples ##

**Example 1**
[Example 1]

**Example 2**
[Example 2]

**Example 3**
[Example 3]

Now, analyze the following prompt:
{prompt_text}
```

**Human Action Detection Prompt**

```
You are an action verification assistant. Your task is to answer questions about
whether a specific action happens in a video. You must always respond strictly with
"Yes" or "No". Do not provide any explanation, reasoning, or additional words. If
you are unsure, answer "No".

Here are some examples as output format reference:

**Example 1**
Question: Did the man run in the video?
Answer: Yes

**Example 2**
Question: Did the woman jump in the video?
```

```
Answer: No

**Example 3**
Question: Does the dog chase a ball in the video?
Answer: Yes

Now analyze the given video:

Question: {question_text}
Answer:
```

**Human Action Smoothness Evaluation Prompt**

```
You are an action smoothness evaluation assistant. Your task is to answer binary
questions about how smoothly an action is performed in a video. You must always
respond strictly with "Yes" or "No". Do not provide any explanation, reasoning, or
additional words. If you are unsure, answer "No".
Now here is the action and question.
Action: {action_text}
Question: {question_text}

# The following part does not occur in original prompts
# Preset questions are as follows:
# [
#     "Was the action continuous without abrupt interruptions?",
#     "Did the action appear natural and not stiff?",
#     "Did the action maintain fluid transitions from start to finish?"
# ]
```

## C.5  CONTENT CLARITY EVALUATION PROMPT

**Content Clarity Evaluation Prompt**

```
You are a **vision-language evaluator**. Your task is to **watch the input video**
and evaluate how well it communicates a coherent and meaningful narrative.

Evaluate across **four dimensions** (score **0-4**):

* **0 = Very Poor**
* **1 = Poor**
* **2 = Acceptable**
* **3 = Good**
* **4 = Excellent**

For each dimension, output a JSON object with:

* `"score"`: the numeric score
* `"reason"`: 1-2 sentences citing what is visible in the video that justifies the
score

**Dimensions:**

1. **Theme Clarity** - Is there a clear central theme or message?
2. **Logical Structure** - Do scenes flow coherently?
3. **Information Completeness** - Is enough visual context provided to understand
the video?
4. **Information Consistency** - Are visual elements consistent across shots?
```

```
Use the following examples only as **format references**. Do **not** align your
scoring with them.

---

### Example 1 (low range)
[Example 1]

### Example 2 (mid range)
[Example 2]

### Example 3 (high range)
[Example 3]
```

## C.6  PROMPTS RELATED TO HERD EVALUATION

**HERD Evaluation Prompt**

```
Please extract some key information from text about someone's feeling after watch a
video and merge these key information into a json format data. I'll give you an
example as format reference. You should consider from the following aspects:

1. Emotional Response: Describe how the video made you feel curious, tense,
inspired, confused, etc.
2. Narrative Flow: Analyze how the story unfolds and whether the pacing feels
smooth or rushed.
3. Character Development: Evaluate how well the characters are developed and how
their relationships evolve.
4. Visual Style: Comment on the use of visuals, color and cinematography to create
atmosphere.
5. Themes: Reflect on the core ideas or social commentary presented in the video.
6. Interpretive Depth: Consider whether the video leaves room for multiple
interpretations or unanswered questions.
7. Overall Impression: Give your overall impression and suggest whether it's worth
watching, and for whom.

If any of them is not mentioned in the input paragraph, you could remain the
default value in the given template.

## Input Text ##
{evaluation_text}
## Ouput Format Reference ##
```json
{
    "Emotional Response": "",
    "Narrative Flow": "",
    "Character Development": "",
    "Visual Style": "",
    "Themes": "",
    "Interpretive Depth": "",
    "Overall Impression": ""
}
```
```

**HERD Questions Generation Prompt**

```
Given a multi-dimensional evaluation of a video, generate one closed-ended question
(answerable with only "yes" or "no") for each dimension.

Each question should:
* Accurately reflect the key message or implication of the original description.
* Use clear and natural phrasing.
* Remain faithful to the tone and nuance of the source content (e.g., themes,
pacing, emotion).
* Be specific enough to elicit a meaningful "yes" or "no" answer.

## Input Format ##
A JSON object, where each key is a dimension and each value is its evaluation.

## Output Format ##
A JSON object where each key is the same dimension, and each value is a yes/no
question derived from the evaluation. Demonstrated as follows:
```json
{
    "Emotional Response": "",
    "Narrative Flow": "",
    "Character Development": "",
    "Visual Style": "",
    "Themes": "",
    "Interpretive Depth": "",
    "Overall Impression": ""
}
```

## Explanation of Each Dimension ##
1. Emotional Response: Describe how the video made you feel curious, tense,
inspired, confused, etc.
2. Narrative Flow: Analyze how the story unfolds and whether the pacing feels
smooth or rushed.
3. Character Development: Evaluate how well the characters are developed and how
their relationships evolve.
4. Visual Style: Comment on the use of visuals, color and cinematography to create
atmosphere.
5. Themes: Reflect on the core ideas or social commentary presented in the video.
6. Interpretive Depth: Consider whether the video leaves room for multiple
interpretations or unanswered questions.
7. Overall Impression: Give your overall impression and suggest whether it's worth
watching, and for whom.

## Input ##
```json
{evaluation_text}
```

## Ouput ##
```

**HERD Questions Polarity Judgment Prompt**

```
You are given a list of evaluative questions about a video.
Each question is designed to check whether the video meets or fails human
expectations in different aspects.
Your task is to classify each question as **positive** or **negative**, based on
the following principle:
```

```
- If the question asks whether the video achieved or matched an intended/expected
effect, then the question is **positive**.
  (In this case, a "Yes" answer indicates the video met expectations.)

- If the question asks whether the video failed to achieve or lacked something that
is expected, then the question is **negative**.
  (In this case, a "Yes" answer indicates the video did not meet expectations.)

Output only "positive" or "negative" for each question.

Question: {question_text}
Answer:
```

# D NUMERICAL RESULTS

## D.1 PERFORMANCE RESULTS OF SAMPLES FROM DIFFERENT THEME CATEGORIES

In order to examine performance variations across different content themes, we summarize the grouped evaluation results in Table 6. Here, prompts are divided into three categories, and the corresponding results across all five dimensions are reported per group, offering further insights into the strengths and weaknesses of each baseline under diverse conditions.

Table 6: Performance results of all baseline methods on five evaluation dimensions. Samples are grouped into three categories according to prompt content themes, and the performance of each group is reported for every method. Note that all values are expressed as percentages to improve readability and conserve space.

| Theme | Methods | Static Quality | Text-Video Alignment | Temporal Quality | HERD | Content Clarity | Avg. |
|---|---|---|---|---|---|---|---|
| Human Real Life | FreeNoise | 68.23 | 54.67 | 74.04 | 51.70 | 72.25 | 64.18 |
| | MEVG | 30.65 | 54.58 | 67.29 | 43.62 | 48.78 | 48.98 |
| | FreeLong | 57.09 | 59.63 | 67.41 | 57.44 | 61.23 | 60.56 |
| | FIFO-Diffusion | 64.15 | 52.20 | 76.52 | 51.97 | 78.95 | 64.76 |
| | DiTCtrl | 59.22 | 62.23 | 70.97 | 57.09 | 61.61 | 62.22 |
| | CausVid | 76.99 | 65.10 | 70.15 | 64.67 | 64.54 | 68.29 |
| | SkyReels-V2 | 77.74 | 59.78 | 80.51 | 64.07 | 75.65 | 71.55 |
| | Vlogger | 65.26 | 44.24 | 66.50 | 59.17 | 53.76 | 57.79 |
| | VGoT | **91.30** | 54.66 | 72.46 | 65.33 | **80.80** | **72.91** |
| | Ground-Truth | 60.94 | **69.08** | 71.25 | **66.88** | 68.81 | 67.39 |
| Nature Exploration | FreeNoise | 69.06 | 59.80 | 72.38 | 56.10 | 75.68 | 66.60 |
| | MEVG | 30.49 | 65.14 | 66.14 | 59.13 | 55.49 | 55.28 |
| | FreeLong | 61.45 | 69.08 | 66.15 | 64.59 | 67.35 | 65.72 |
| | FIFO-Diffusion | 65.32 | 55.77 | 74.49 | 55.41 | 83.56 | 66.91 |
| | DiTCtrl | 66.43 | 71.40 | 70.32 | 65.97 | 68.98 | 68.62 |
| | CausVid | 78.00 | 72.77 | 69.58 | 68.62 | 69.58 | 71.71 |
| | SkyReels-V2 | 75.76 | 60.57 | 79.42 | 66.06 | 74.47 | 71.26 |
| | Vlogger | 63.20 | 52.23 | 66.74 | 65.84 | 52.16 | 60.03 |
| | VGoT | **91.67** | 56.30 | 70.06 | 68.48 | **85.08** | **74.32** |
| | Ground-Truth | 68.06 | **75.25** | 69.20 | 68.24 | 66.84 | 69.51 |
| Virtual Entertainment | FreeNoise | 67.24 | 51.39 | 73.06 | 40.35 | 74.78 | 61.36 |
| | MEVG | 28.23 | 53.51 | 66.42 | 43.82 | 40.20 | 46.44 |
| | FreeLong | 52.48 | 56.02 | 65.73 | 56.69 | 53.66 | 56.92 |
| | FIFO-Diffusion | 61.79 | 48.71 | 75.37 | 39.90 | 78.94 | 60.74 |
| | DiTCtrl | 50.39 | 58.94 | 70.97 | 63.45 | 58.90 | 60.13 |
| | CausVid | 73.86 | **63.75** | 69.63 | 59.97 | 50.84 | 63.61 |
| | SkyReels-V2 | 66.13 | 55.76 | **77.99** | 61.26 | 71.17 | 66.46 |
| | Vlogger | 62.61 | 39.75 | 64.63 | 53.67 | 38.96 | 51.12 |
| | VGoT | **89.52** | 56.35 | 70.60 | 60.75 | 74.72 | **70.39** |
| | Ground-Truth | 54.37 | 59.77 | 70.37 | **65.49** | 65.19 | 63.03 |

# E  CASE STUDY

## E.1  EXAMPLES GENERATED BY BASELINE METHODS

We provide some evaluation cases as follows to demonstrate how our evaluation framework perform on test samples and compare the capabilities of different methods in a more perceptive aspect. Specifically, we illustrate the prompt base (i.e. description text to derive the prompt mentioned in Section 3.1), scores in five major evaluation dimensions and frame sequence in which frames are uniformly sampled from corresponding generated videos. The chosen samples are "food_3" generated by FIFO-Diffusion (Kim et al., 2024) and DiTCtrl (Cai et al., 2025) (Fig. 11), "minivlog_9" generated by VGoT (Zheng et al., 2024) and MEVG (Oh et al., 2024) (Fig. 12) and "pets_8" generated by CausVid (Yin et al., 2025) and SkyReels-V2 (Chen et al., 2025a) (Fig. 13). In this way we might grasp more explicit understanding of our evaluation framework mechanism and intuitively feel the gaps between the videos generated by different models.

## E.2  EXAMPLES GENERATED BY PROPRIETARY METHODS

Proprietary models such as Gen4 (Runway AI, 2025), Veo3 (DeepMind, 2025), and Kling-2.0 (Kuaishou, 2025) currently represent the state of the art in video generation. However, these models still struggle to produce long-form videos with multiple scenes, especially those exceeding 30 seconds. To evaluate the performance of these powerful proprietary systems on our LoCoT2V-Bench, we design a workflow that integrates story visualization models with closed-source Image-to-Video (I2V) models to generate the target videos. Following (Shi et al., 2025), we adopt Story-Adapter (Mao et al., 2024) together with Seedance-1.0-pro (Gao et al., 2025) to construct this pipeline, which we refer to as **SA-SD** for brevity.

Although the constructed workflow can demonstrate the performance of proprietary models on our benchmark to some extent, it incurs substantial financial costs. Specifically, even though LoCoT2V-Bench contains only 240 test samples, using the event information in the prompt suite requires generating more than 1700 video clips, each of which is expensive. Therefore, instead of reporting full benchmark results, we present the scores of selected examples produced by SA-SD. The IDs of chosen samples are "film_17" (Fig. 14), "soap_opera_4" (Fig. 15) and "wildlife_3" (Fig. 16). These examples offer a partial yet informative glimpse into the capabilities of proprietary models.

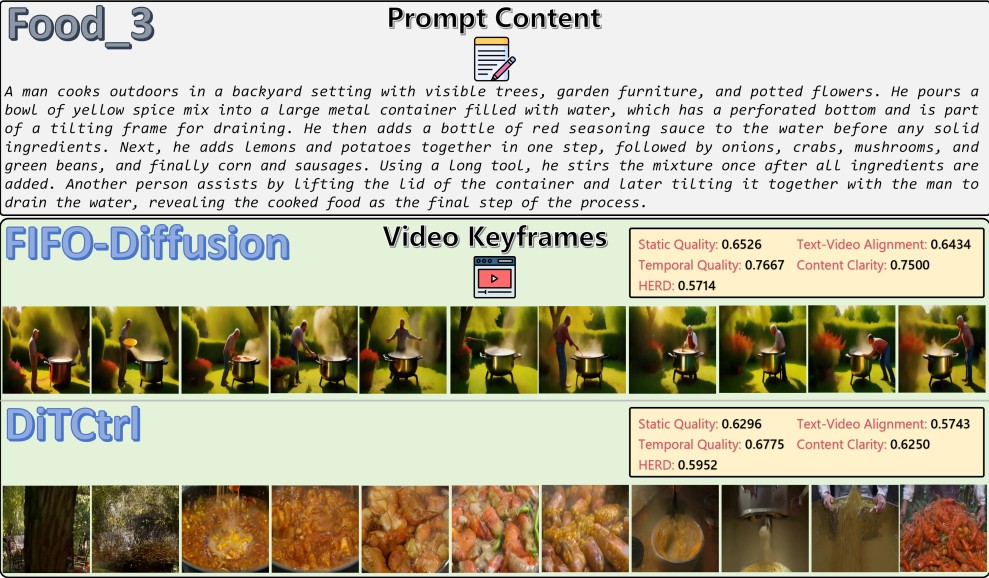

Figure 11: **Case 1.** Evaluation sample "food_3" of generated by FIFO-Diffusion and DiTCtrl.

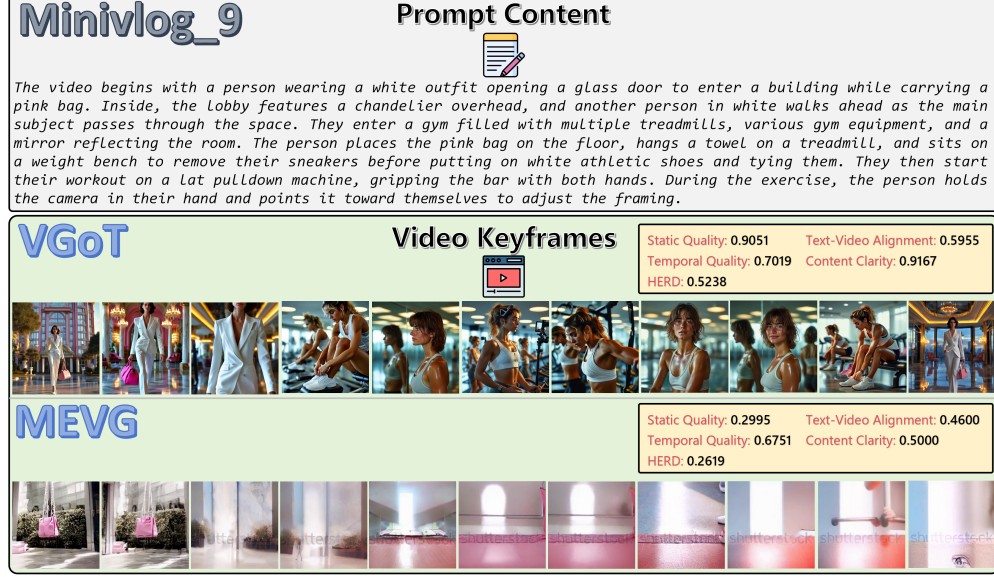

Figure 12: **Case 2.** Evaluation sample of "minivlog_9" generated by VGoT and MEVG.

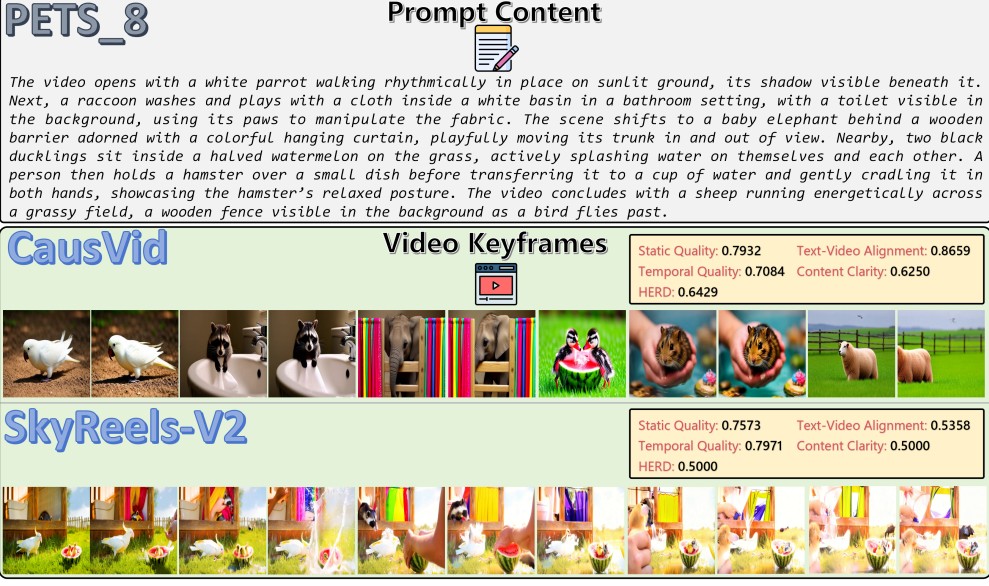

Figure 13: **Case 3.** Evaluation sample of "pets_8" generated by CausVid and SkyReels-V2.

## F  DETAILED LLM USAGE IN OUR WORK

We mainly use LLMs to support our writing process. Specifically, we first draft text manually and then prompt LLMs to help polish the language. We carefully check and revise any unexpected content generated by the models, sometimes through multiple rounds of refinement. In addition, we use LLMs to assist in generating complex LaTeX code, particularly for displaying intricate plots or tables. These constitute the primary ways in which we employ LLMs.

**Film_17** **Prompt Content**

*The video begins with a car driving past a house, seen from the interior of a moving vehicle with a visible side mirror. Two children—a boy and a girl—are seen riding scooters nearby, approaching a different house with a police car parked outside. The scene then shifts to Maybrook Elementary School, where students exit a yellow school bus in an overhead shot and walk into the building. Inside the school, students move through the halls and enter classrooms. A teacher walks into one of the classrooms, carrying a bag over one shoulder and holding a clipboard in one hand. Later, at night, a clock on a bedside table in a dark bedroom shows 2:17 AM as someone (not visible) sneaks out of the house; the movement of the bedcovers and the opening of the front door are shown. The static camera captures the staircase and front door from a fixed position as the unseen person descends and steps outside.*

**SA-SD** **Video Keyframes**

Static Quality: **0.7805**   Text-Video Alignment: **0.6460**
Temporal Quality: **0.6807**   Content Clarity: **0.7500**
HERD: **0.5476**

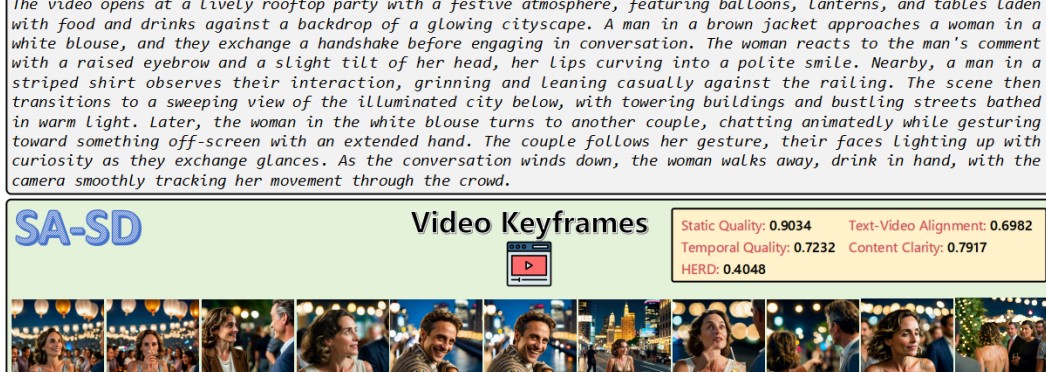

Figure 14: **Case 4.** Evaluation sample of "film_17" generated by SA-SD.

**Soap_Opera_4** **Prompt Content**

*The video opens at a lively rooftop party with a festive atmosphere, featuring balloons, lanterns, and tables laden with food and drinks against a backdrop of a glowing cityscape. A man in a brown jacket approaches a woman in a white blouse, and they exchange a handshake before engaging in conversation. The woman reacts to the man's comment with a raised eyebrow and a slight tilt of her head, her lips curving into a polite smile. Nearby, a man in a striped shirt observes their interaction, grinning and leaning casually against the railing. The scene then transitions to a sweeping view of the illuminated city below, with towering buildings and bustling streets bathed in warm light. Later, the woman in the white blouse turns to another couple, chatting animatedly while gesturing toward something off-screen with an extended hand. The couple follows her gesture, their faces lighting up with curiosity as they exchange glances. As the conversation winds down, the woman walks away, drink in hand, with the camera smoothly tracking her movement through the crowd.*

**SA-SD** **Video Keyframes**

Static Quality: **0.9034**   Text-Video Alignment: **0.6982**
Temporal Quality: **0.7232**   Content Clarity: **0.7917**
HERD: **0.4048**

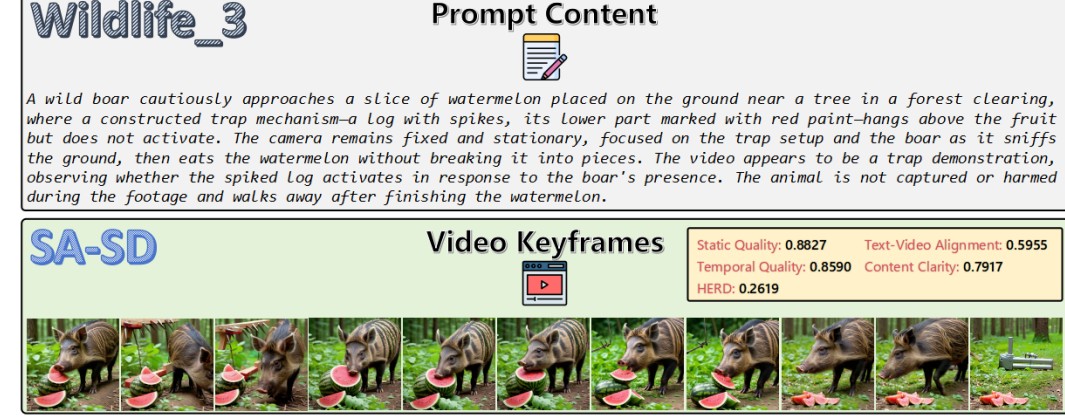

Figure 15: **Case 5.** Evaluation sample of "soap_opera_4" generated by SA-SD.

**Wildlife_3** **Prompt Content**

*A wild boar cautiously approaches a slice of watermelon placed on the ground near a tree in a forest clearing, where a constructed trap mechanism—a log with spikes, its lower part marked with red paint—hangs above the fruit but does not activate. The camera remains fixed and stationary, focused on the trap setup and the boar as it sniffs the ground, then eats the watermelon without breaking it into pieces. The video appears to be a trap demonstration, observing whether the spiked log activates in response to the boar's presence. The animal is not captured or harmed during the footage and walks away after finishing the watermelon.*

**SA-SD** **Video Keyframes**

Static Quality: **0.8827**   Text-Video Alignment: **0.5955**
Temporal Quality: **0.8590**   Content Clarity: **0.7917**
HERD: **0.2619**

Figure 16: **Case 6.** Evaluation sample of "wildlife_3" generated by SA-SD.

