# OpenReview forum: "LoCoT2V-Bench: A Benchmark for Long-Form and Complex Text-to-Video Generation"
_ICLR.cc/2026/Conference — Submitted to ICLR 2026_

### Official Review · Reviewer_RBq7 · 2025-10-21

**Soundness:** 3
**Presentation:** 3
**Contribution:** 3
**Rating:** 6
**Confidence:** 4

**Summary:**

This paper introduces LoCoT2V-Bench, a new benchmark designed specifically for evaluating long video generation under complex input conditions.
The authors argue that existing benchmarks often rely on simplified prompts and focus on low-level metrics, neglecting higher-level dimensions such as narrative coherence and thematic expression. To address this gap, LoCoT2V-Bench presents two main contributions: (1) A suite of longer and more complex prompts derived from real-world videos, incorporating elements like scene transitions and event dynamics. (2) A multi-dimensional evaluation framework that, in addition to traditional metrics, proposes novel dimensions such as event-level alignment, content clarity, and HERD.

**Strengths:**

**1. Well-motivated and Significant Problem**

As video generation models advance, the evaluation of longer, more complex videos has become a critical challenge. This paper accurately identifies the limitations of existing benchmarks and proposes a solution tailored for long video under complex prompts, which is crucial for advancing the field.

**2. Comprehensive Evaluation Dimensions**

The evaluation is very comprehensive.  It not only covers fundamental metrics like Static Quality, Text-Video Alignment, and Temporal Quality but also innovatively introduces higher-level dimensions like Content Clarity and the Human Expectation Realization Degree. HERD, in particular, is a valuable exploration in video generation evaluation as it attempts to quantify abstract concepts like emotion, narrative, and character development.

**3. Extensive Experiments and In-depth Analysis**

The authors have conducted a thorough evaluation of existing open-source LVG models. The evaluation not only reports overall performance but also delves into deeper exploration (e.g. Sec 4.2~4.4).

**Weaknesses:**

**1. Limited Scale of the Benchmark**

The benchmark consists of 240 samples distributed across 18 themes. My main concern is that this sample size may be insufficient to draw robust evaluation about model capabilities.

**2. Evaluation Reliability of HERD**

The proposed automated evaluation pipeline, especially the HERD metric, is heavily dependent on multiple third-party LLM/MLLMs. The generation and evaluation process for HERD involves a long chain of model calls. How do the authors evaluate and control the accumulated errors in this chain? How sensitive is the final metric to a potential failure at any stage of this pipeline?

**3. Handling the Errors during Evaluation**

For fine-grained metrics like "event-level temporal consistency," the calculation seems to presuppose that the corresponding events and subjects can be accurately located in the video. I think this assumption is questionable in complex scenarios. Also, there might be multiple subjects in the same video clip, or even in the same video frame. Does subject extraction confuse different subjects in these complex settings?
It is also unclear how the system robustly handles intermittent subject presence, such as when a subject is occluded or temporarily leaves the frame and then reappears.

**Questions:**

Please see weaknesses.

---

> ### Author Response · Authors · 2025-12-02
> **Response to Reviewer RBq7**
>
> We greatly appreciate the reviewer’s insightful observations and constructive critiques. A point-by-point response is provided below.

---

> ### Author Response · Authors · 2025-12-02
> **Response to Reviewer RBq7**
>
> > **W1**: **Limited Scale of the Benchmark**: The benchmark consists of 240 samples distributed across 18 themes. My main concern is that this sample size may be insufficient to draw robust evaluation about model capabilities.
>
> **R1**: We acknowledge that a larger benchmark can further strengthen the statistical robustness of the evaluation. However, the following issues are also supposed to be considered:
>
> 1. **Expanding the scale of long-video benchmarks poses several practical challenges.** First, as mentioned in collecting suitable source videos across the 18 themes is highly non-trivial: even on large platforms such as YouTube, videos that simultaneously satisfy our stringent content and structural requirements are rare, and the overall quality of crawled videos varies significantly. We employed multiple filtering strategies and strict quality control to ensure that the selected videos are clean, diverse, and representative, which substantially limits the feasible scale.
> 2. **Generating long videos incurs high computational and, for closed-source systems, financial cost.**  Increasing the dataset size would significantly amplify these burdens, as long video generation is a task with high computing demands. While scaling the benchmark to thousands of samples would indeed yield more robust statistics, it would also make the evaluation prohibitively expensive. For example, evaluating **only SkyReels-V2** on our existing 240-sample benchmark requires **approximately 480 hours** (using **two A100 80G GPUs**). Scaling this to over a thousand samples would result in multi-week evaluation time and significantly higher computational cost, making the benchmark impractical for most research groups.
>
> For these reasons, we chose a moderate scale that balances diversity, quality, and practical evaluability. We believe the curated 240-sample benchmark provides a meaningful and reproducible assessment of long-video generation capabilities while remaining computationally feasible.
>
> >  **W2**: **Evaluation Reliability of HERD**: The proposed automated evaluation pipeline, especially the HERD metric, is heavily dependent on multiple third-party LLM/MLLMs. The generation and evaluation process for HERD involves a long chain of model calls. How do the authors evaluate and control the accumulated errors in this chain? How sensitive is the final metric to a potential failure at any stage of this pipeline?
>
> **R2**: We fully understand the reviewer’s concerns regarding reliability and error accumulation within the HERD pipeline. To mitigate these issues, we adopt the following measures:
>
> 1. **Human verification to ensure faithful interpretation of HERD dimensions**: Although the third-party LLM/MLLMs we employ are strong, they may occasionally misinterpret the definition of certain HERD dimensions (e.g., narrative flow, character development) or the polarity of specific questions. To prevent such errors from propagating, we manually inspect and refine all LLM-generated HERD evaluations and questions sample by sample, ensuring that every item faithfully reflects our intended semantics.
> 2. **Control of accumulated errors in the multi-stage pipeline**: Each intermediate output in the pipeline is validated before being used in the next stage. This staged verification prevents errors at one step from silently influencing subsequent steps. In other words, the pipeline is not executed blindly end-to-end; it is gated by human checks at critical semantic points.
> 3. **Stability of final scores**: We have also observed that the final HERD scores are relatively insensitive to small variations in intermediate LLM outputs due to our manual check and VQA methods with multiple questions for each dimension ensuring both the content quality and robustness to potential issues like outputs' randomness or LLM's hallucinations.

---

> ### Author Response · Authors · 2025-12-02
> **Response to Reviewer RBq7**
>
> > **W3**: **Handling the Errors during Evaluation**: For fine-grained metrics like "event-level temporal consistency," the calculation seems to presuppose that the corresponding events and subjects can be accurately located in the video. I think this assumption is questionable in complex scenarios. Also, there might be multiple subjects in the same video clip, or even in the same video frame. Does subject extraction confuse different subjects in these complex settings? It is also unclear how the system robustly handles intermittent subject presence, such as when a subject is occluded or temporarily leaves the frame and then reappears.
>
> **R3**: We really agree that relying on event and subject localization introduces challenges in complex scenarios. To mitigate these issues, we adopted the following strategies:
>
> 1. **Use of powerful vision models**: We intentionally employ state-of-the-art models for the required sub-tasks—e.g., Time-R1 for temporal event parsing and Grounding-SAM2 for subject grounding—which demonstrate strong performance in their respective domains. While no model can perfectly handle all out-of-domain or highly complex cases, we observe that these tools perform stably and reliably for the types of scenes included in our benchmark.
> 2. **Handling Intermittent Subject Presence**: Our metric considers only frames in which the subject is visible. Occluded or absent subjects pose no consistency requirement, as there is no visual evidence to compare. The evaluation focuses on ensuring that *whenever* the same subject reappears, its appearance remains stable across frames.
> 3. **Subject Disambiguation in Multi-Subject Scenes**: Potential confusion between multiple subjects largely depends on the grounding model’s capability. Grounding-SAM2 outputs all candidate segments rather than a single matched instance and maintains correspondence across frames. Combined with the subject descriptors extracted from LLM-generated prompts—which provide complete and unambiguous subject attributes—we compute consistency for all grounded subject instances. Therefore, even if multiple subjects are present, the metric remains robust and is not affected by subject co-occurrence.
>
> Overall, while we acknowledge that extremely complex edge cases may be challenging for any automatic system, the chosen models and evaluation design ensure stable and reliable behavior within the scope of our benchmark.

---

### Official Review · Reviewer_mQKm · 2025-10-28

**Soundness:** 2
**Presentation:** 3
**Contribution:** 3
**Rating:** 6
**Confidence:** 4

**Summary:**

This paper presents LoCoT2VBench, a benchmark for evaluating video generative models on long video generation. It presents a prompt suite consisting of 240 samples from 18 themes, derived from real-world videos, and an evaluation framework consisting of a taxonomy of novel metrics.

The paper also evaluates open source video generation methods on these metrics, and does an insightful analysis over key questions derived from them, like the correlation of video content type and evaluation results, relationship between static metrics (like image quality) and the long video ones, and the impact of prompt complexity in the results.

Prompt data, benchmark results, and evaluation code will be released with the paper (the first two having already been published anonymously).

**Strengths:**

The following are some of the key strengths of this paper:

* The problem this paper focuses on is indeed important, and increasingly so as the video generation models improve
* The taxonomy of metrics to analyze is very well selected, thoughtful, and exhaustive. Definitely a strong point for this work. Particularly the approach for event-level alignment is interesting; but overall many of these metrics are definitely insightful
* The questions stated in the Results & Analysis section are interesting and the analysis well done. Even (perhaps) more intuitive results like "methods tend to perform worse with higher semantic complexity, indicating their difficulty in understanding complex semantics" are made insightful by the quantitative approach.
* This work leverages the strengths of modern Vision-Language Models correctly
* The code being open source will be great for reproducibility, once it's made available
* Good ethics section

**Weaknesses:**

These are the main weaknesses of this work:

* Primarily, there isn't a comparison against human evaluation, so it's unclear how these metrics compare to reality. This is the main point that, if fixed, would greatly enhance this work.
  * As an example of how this matters, it is claimed that for the Overall Alignment metric, existing benchmarks use CLIP-based computations but for this work an approach based on MLLMs summarization followed by semantic similarity is preferable. But no quantitative comparison between both approaches is made; if there was a measure of human alignment, this claim could be verified.
  * In another point in the paper, it is concluded that "the differences across theme categories are minor, which (...) demonstrates the robustness of our evaluation framework to diverse content types". I don't think this claim is guaranteed without a human baseline.

* Similarly, the benchmark is only applied to the output of generative video models, but never to ground truth data. If the authors applied the benchmark to real videos one could know what the upper bound of quality is, and it would make all analysis and plots more complete.

Other misc weaknesses:

* All baselines are open source; including at least some proprietary baselines would increase the variety and provide some additional insights (e.g. what is the gap, if any, between open and closed source systems with respect to these metrics?)
* The source is not currently available (hopefully it can be made available before the end of the review period)

Other smaller issues regarding clarity:

* Early on, the paper introduces 7 dimensions in the "Evaluation Information Integration" section. Later on, the paper starts describing a more complex taxonomy of metrics split into 5 groups (in the "Evaluation Dimension Suite Construction" section). 2-3 pages later the proper context for the 7 dimensions (HERD) is introduced; this makes the work harder to follow.

* Some phrasing is ambiguous, particularly "existing LVG methods exhibit clear improvements in frame-level static quality" (what is the improvement compared against?). A few typos are also present (I can now recall "adpot")

**Questions:**

Most questions raised during the review are expressed in the Weaknesses section; a few less important considerations that I'd be interested in hearing about in the comments section of this review:

* Have the authors thought about possible extensions to video + audio?
* What insights (if any) do you derive from these results, in terms of how to improve video models?

---

> ### Author Response · Authors · 2025-12-02
> **Response to Reviewer mQKm**
>
> We appreciate the reviewer’s valuable input and the effort devoted to assessing our submission. We respond to each point in detail below.

---

> ### Author Response · Authors · 2025-12-02
> **Response to Reviewer mQKm**
>
> > **W2**: Similarly, the benchmark is only applied to the output of generative video models, but never to ground truth data. If the authors applied the benchmark to real videos one could know what the upper bound of quality is, and it would make all analysis and plots more complete.
>
> **R1**: We appreciate the reviewer’s suggestion and have added the corresponding analysis in the updated rebuttal materials. We would like to note, however, that ground-truth videos should not be interpreted as an upper bound within our evaluation framework. In our benchmark design, these videos are used to derive prompts rather than to define target outputs that models are expected to reproduce. Even when enriched using strong MLLMs/LLMs—along with HERD-based descriptions of the ground-truth content—the prompts do not obligate generated videos to match the original footage. As a result, high scores can be assigned not because a model mimics the ground-truth video, but because it produces a high-quality, prompt-consistent multi-scene video. Thus, while informative, ground-truth performance does not inherently represent the ceiling of our metric space.
>
> > **W3**: All baselines are open source; including at least some proprietary baselines would increase the variety and provide some additional insights (e.g. what is the gap, if any, between open and closed source systems with respect to these metrics?)
>
> **R2**: We agree that including proprietary baselines could be informative. However, for long, multi-scene video generation there is currently no widely accepted, reliable benchmark for video quality; most prior work reports performance through qualitative case studies rather than standardized, comparable metrics, which makes it difficult to single out a clear SOTA system. In addition, the majority of commercial text-to-video APIs we surveyed only support very short (5–10s), single-scene clips with restricted content, and we could not identify any closed-source text-to-video service capable of directly producing the 30–60s, multi-scene outputs required by our benchmark.
>
> To nevertheless provide a view into closed-source performance, we evaluated an alternative pipeline that stitches together storyboard/keyframe generation with proprietary image-to-video components. Concretely, we used the *Story-Adapter* [1] in combination with a closed-source I2V model *seedance-pro-1.0* [2]. Because calling these proprietary services is both computationally and financially expensive, we ran only three examples and present them as illustrative case studies in the appendix. The manuscript has been updated to include these additional comparisons and a brief discussion of their limitations.
>
> ---
>
> [1] Mao J et al. Story-Adapter: A Training-free Iterative Framework for Long Story Visualization. arXiv:2410.06244
>
> [2] Gao Y et al. Seedance 1.0: Exploring the Boundaries of Video Generation Models. arXiv:2506.09113
>
> > **W4**: The source is not currently available (hopefully it can be made available before the end of the review period)
>
> **R3**: We are fully committed to releasing all benchmark resources (code, data-processing scripts, and prompts). The materials are currently being cleaned and organized to ensure reproducibility, and therefore cannot be released immediately. We are actively preparing the repository and will make it publicly available as soon as it is ready.
>
> > **W5**: Early on, the paper introduces 7 dimensions in the "Evaluation Information Integration" section. Later on, the paper starts describing a more complex taxonomy of metrics split into 5 groups (in the "Evaluation Dimension Suite Construction" section). 2-3 pages later the proper context for the 7 dimensions (HERD) is introduced; this makes the work harder to follow.
>
> **R4**: Thanks for pointing out this ambiguity. We've refined the expression of the corresponding part and stress it in green font. Please see it in our new submission (Section 3.1 "Evaluation Information Integration" Part).
>
> > **W6**: Some phrasing is ambiguous, particularly "existing LVG methods exhibit clear improvements in frame-level static quality" (what is the improvement compared against?). A few typos are also present (I can now recall "adpot")
>
> **R5**: We really appreciate the reviewer for pointing out these typo errors. We agree that the phrasing is ambiguous. We will revise the statement to clearly specify what the improvements are compared against and ensure that the description precisely reflects our intended meaning. We will also carefully correct the typo errors like “adpot” in the revised manuscript. Please see them in our new submission.

---

> ### Author Response · Authors · 2025-12-02
> **Response to Reviewer mQKm**
>
> > **Q1**: Have the authors thought about possible extensions to video + audio?
>
> **R6**: Thanks for the insightful question. We address it from the following perspective:
>
> 1. **We did consider extending the evaluation to the “video + audio” setting.** However, most existing long-video generation methods do not support audio generation. Among all baseline models we tested, only Vlogger produces audio–visual outputs, and its audio is limited to voice-over narration of the visual content. As a result, even if we designed audio-related evaluation metrics, a meaningful comparison would not be possible because other methods do not generate audio at all. Hopefully, we would like to extend our prompt suite and evaluation suite to "video+audio" version if more excellent works that support longer videos generation with audio emerged.
> 2. **We agree that extending the benchmark to the “video + audio” setting is an important direction for future work.** Incorporating audio introduces the need to assess alignment across text, video, and audio modalities. A careful evaluation would require verifying both content-level consistency (e.g., whether an explosion mentioned in the prompt is reflected in both the visuals and the sound) and temporal synchrony, ensuring that events occur at matching times across modalities. We believe that establishing reliable protocols for such multimodal evaluation will require additional focused research, and we are very open to exploring these extensions in future versions of the benchmark.
>
> > **Q2**: What insights (if any) do you derive from these results, in terms of how to improve video models?
>
> **R7**: We thank the reviewer for this intriguing and thoughtful question. We are willing to share our views on this issue and we think the following parts may be promising directions for future video generation models:
>
> 1. **Better understanding for more complex inputs**: Our results reveal substantial limitations in both overall and event-level text–video alignment (Table 2), indicating that current models struggle to encode and translate complex, multi-event prompts into coherent video content. We attribute this to their inadequate understanding of complex prompts. Given recent success in unified models, we suppose enhancing the comprehension capability to the inputs may help for generating videos that are more aligned to complex inputs.
> 2. **Augment high-level semantic supervision**: The HERD evaluation reveals that current models fail to meet human expectations in abstract dimensions: *narrative flow* (average score ~28%), *character development* (\~35%), and *interpretive depth* (\~37%) (as shown in Table 4). These gaps persist even when visual quality is high, proving that pure visual optimization is insufficient. We think incorporating more high-level semantic supervision during training may be helpful. (Certainly how to incorporate these kinds of supervision is also an important issue.)
> 3. **Enhance long-range and cross-scene temporal consistency**: Our results highlight a stark contrast between *intra-event* and *inter-event* consistency: while models perform well on intra-event SC/BC, their inter-event SC and BC scores are much lower. This highlights a core challenge in sustaining long-range visual coherence over extended temporal horizons—a difficulty commonly noted by both researchers and practitioners. We hypothesize that approaches grounded in autoregressive modeling may offer a path forward, despite their own limitations in handling very long dependencies. And in our opinions, how to maintain intermediate representations of certain entities and make them serve as conditions for subsequent frames generation should also be attached great importance.

---

### Official Review · Reviewer_hFUp · 2025-11-01

**Soundness:** 3
**Presentation:** 3
**Contribution:** 3
**Rating:** 4
**Confidence:** 5

**Summary:**

This paper introduces LoCoT2V-Bench, a new benchmark specifically designed to evaluate long-form and complex text-to-video (T2V) generation , aiming to address the limitations of existing benchmarks that focus on short clips and simple prompts. The work constructs a challenging prompt suite of 240 samples with an average length of 236 words , derived from real-world YouTube videos. Its core contribution is a multi-dimensional evaluation framework that, beyond static and temporal quality, introduces novel metrics: Event-level Alignment to assess the accurate sequencing of multiple events; Content Clarity to evaluate narrative logic and thematic coherence using MLLMs ; and the Human Expectation Realization Degree (HERD) to quantify abstract aspects like emotional response and narrative flow . By testing nine representative models , the paper finds that while current methods perform adequately in basic visual fidelity , they struggle significantly with fine-grained event alignment, long-range temporal consistency, and high-level narrative adherence .

**Strengths:**

1. It introduces LoCoT2V-Bench, the first systematic benchmark specifically designed for evaluating long-form and complex text-to-video (LVG) generation, addressing a critical gap left by existing benchmarks that primarily focus on short clips and simple prompts.
2. It proposes a novel suite of high-level evaluation dimensions to assess abstract narrative aspects often overlooked by traditional metrics. Key innovations includes Human Expectation Realization Degree, Event-level Alignment and Content Clarity.

**Weaknesses:**

1. Lack of Human Validation: The paper heavily relies on MLLMs to score subjective metrics like HERD but provides no meta-evaluation to correlate these automated scores with actual human judgments, leaving the validity and reliability of these new metrics unconfirmed.
2. Limited Dynamics Assessment: The paper's "Temporal Quality" evaluation focuses primarily on frame-level dynamics (e.g., Motion Smoothness) , while overlooking higher-level dynamics such as the inter-segment and video-level dynamics defined in [1]. This raises the question of whether changing the methods for measuring dynamics would affect the conclusions.
3. Limited Model Scope: The evaluation is restricted to nine open-source models , excluding state-of-the-art (SOTA) closed-source models. This limits the representativeness of the paper's conclusions that current models universally struggle with these complex tasks .


[1] Evaluation of text-to-video generation models: A dynamics perspective.

**Questions:**

see weakness above. I would raise the score if the author can solve my concerns.

---

> ### Author Response · Authors · 2025-12-02
> **Response to Reviewer hFUp**
>
> We thank you for your careful evaluation of our manuscript and your constructive comments. Our point-by-point responses are provided below.

---

> ### Author Response · Authors · 2025-12-02
> **Response to Reviewer hFUp**
>
> > **W2**: Limited Dynamics Assessment: The paper's "Temporal Quality" evaluation focuses primarily on frame-level dynamics (e.g., Motion Smoothness) , while overlooking higher-level dynamics such as the inter-segment and video-level dynamics defined in [1]. This raises the question of whether changing the methods for measuring dynamics would affect the conclusions.
>
> > [1] Evaluation of text-to-video generation models: A dynamics perspective.
>
> **R1**: We thank the reviewer for the insightful comment and for highlighting the potential limitations of our current dynamics assessment. We would like to clarify the following points:
>
> 1. **Our “Temporal Quality” assessment is not limited to frame-level dynamics**: The goal of this component is to evaluate whether the generated video successfully conveys the dynamic elements implied by the prompt. For instance, the “human action” metric assesses whether the action semantics are correctly demonstrated. While several metrics rely on frame-level similarity measurements (e.g., Inter-event Subject Consistency), this does not imply that our evaluation focuses solely on low-level dynamics; rather, these computations serve as proxies for capturing dynamic semantic fidelity.
> 2. **Fine-grained (frame-level) dynamics are often more error-prone and perceptually impactful**: In practice, subtle temporal distortions—such as jitter, temporal inconsistency, or incorrect short-term transitions—tend to significantly degrade perceived video quality. Therefore, our benchmark currently prioritizes metrics that are sensitive to such fine-grained temporal issues. Nevertheless, we agree that incorporating coarser, higher-level dynamic assessments would provide a more comprehensive evaluation, and we intend to explore this direction in future benchmark extensions.
> 3. **High-level dynamics metrics are still underdefined and unstable in current research practice**: Although we acknowledge the importance of higher-level dynamics (e.g., inter-segment or event-level coherence), existing methods for measuring them often suffer from inconsistent definitions, task-dependent assumptions, and large variance across models and prompts. Without reliable semantic boundary detection, these metrics may introduce additional noise rather than provide meaningful discrimination. This is one of the reasons we currently focus on metrics with clearer definitions and more stable measurement properties.
> 4. **Video segmentation for higher-level dynamic assessment remains an open challenge**: Defining meaningful temporal segments is non-trivial: segmenting videos into equal-length intervals may not align with semantic boundaries, and automatic segmentation that respects event-level coherence remains an active research problem. We believe that establishing robust segmentation strategies is a prerequisite for introducing reliable higher-level dynamic metrics.
> 5. **Our benchmark design is modular and allows easy extension to incorporate hierarchical dynamics**: Importantly, the evaluation framework is designed to be extensible. Metrics operating at higher granularity levels—such as those proposed in [1]—can be integrated once mature segmentation or event-boundary detection methods become available. We see this as a promising direction and plan to expand our benchmark along this axis in future versions to provide a more comprehensive multi-scale dynamics evaluation.
>
> > **W3**: Limited Model Scope: The evaluation is restricted to nine open-source models , excluding state-of-the-art (SOTA) closed-source models. This limits the representativeness of the paper's conclusions that current models universally struggle with these complex tasks.
>
> **R2**: We agree that incorporating closed-source SOTA models would further strengthen the evaluation. However, long video generation—especially multi-scene generation—still lacks unified and reliable benchmarks, and most commercial APIs currently support only short (5–10 s) single-scene T2V outputs, making them incompatible with our task setting. Moreover, we were unable to identify any proprietary T2V system that can directly produce the 30–60s multi-scene videos required by our benchmark.
>
> To address this concern, we explored an alternative approach to include closed-source models: we constructed a pipeline that combines storyboard-based keyframe generation with a proprietary Image-to-Video system. Specifically, we employed *story-adapter* [1] together with the commercial I2V model *seedance-pro-1.0* [2]. Due to the significant computational and financial cost of these services, we evaluated three representative examples and reported them as case studies in the appendix. The updated manuscript now includes these additional comparisons.
>
> ---
> [1] Mao J et al. Story-Adapter: A Training-free Iterative Framework for Long Story Visualization. arXiv:2410.06244
>
> [2] Gao Y et al. Seedance 1.0: Exploring the Boundaries of Video Generation Models. arXiv:2506.09113

---

### Official Review · Reviewer_f6iZ · 2025-11-02

**Soundness:** 2
**Presentation:** 2
**Contribution:** 2
**Rating:** 2
**Confidence:** 4

**Summary:**

This paper introduces a new benchmark for long video generation (30–60s). The dataset consists of 240 text prompts, obtained by captioning real videos with SOTA VLMs. The authors also propose an evaluation suite across five categories. Experiments on 9 open models suggest that current models struggle with high-level adherence and temporal consistency.

While the topic is very relevant to the community, I believe the paper needs a thorough verification of the proposed metrics as well as improved writing of the dataset and metrics. Because of these reasons, I do not think the paper is ready for publication at this time.

**Strengths:**

1. A new prompt set for long video generation, with longer and more complex prompts generated by VLMs and verified by humans.
2. An evaluation suite that includes previously used dimensions (e.g. quality and text alignment) as well as high-level dimensions such as emotional response.
3. Experiments with 9 different models, including analyses of entanglement between different dimensions.

**Weaknesses:**

1. For a paper presenting a new dataset, none of the actual prompts (and corresponding real videos) are shown in the main paper. This makes it difficult to understand the data creation and the complexity of the prompts from the main paper alone. I only saw 3 prompts in Appendix E.
2. Details and takeaways from the prompt generation are missing. It would be informative to know how each of the stages (raw captioning, self-refine, manual review) change the prompt set, so the community can decide whether and how to build on top of this pipeline.
3. A large portion of the paper is used to introduce a large number of metrics to measure different aspects of generation. Nevertheless, some of the metrics are not clearly defined, entirely referring the reader to other papers (eg. L281). While details can be read in other papers, the main body of this paper should be self-contained, providing the minimum details needed for the reader to understand the methodology.
4. Importantly, there is no correlation analysis for any of the proposed metrics to verify that they measure what the authors intended. This is important not only for newly proposed metrics but also for existing metrics, as they are applied to a different setting (long videos) than initially proposed. This casts doubts on all the results and conclusions from the paper.
5. The number of metrics is also substantial, making it hard to process the different results. While more metrics can capture more nuances, it is important for a benchmark to provide a clear (sub)set of metrics to drive progress in the area.
6. Related to the point above, some metrics are related to each other (L412-413). Therefore, it should be verified (with human evaluations) whether they actually capture different aspects, or otherwise be removed from the evaluation suite.
7. Some of the new metrics, such as narrative flow and character development seem too ambitious for 30-60s videos. It would be great to have a human study to identify whether (or for which prompts) they are indeed meaningful.
8. It would also be great to measure how the real, ground-truth videos fare with respect to the metrics. This could be seen as some kind of upper-bound, which would increase credibility in the results.
9. Some older yet relevant related work is missing. This includes models such as Phenaki (Villegas et al., ICLR’23), which was the first model for multi-event generation, and benchmarks such as StoryBench (Bugliarello et al., NeurIPS’23) which evaluates multi-prompt generations and shows that metrics like DOVER do not correlate well with human preferences.


---
Villegas et al. Phenaki: Variable Length Video Generation From Open Domain Textual Description. ICLR’23

Bugliarello et al. StoryBench: A Multifaceted Benchmark for Continuous Story Visualization. NeurIPS’23

**Questions:**

1. It would be nice to define early on in the paper what “long” video generation means (i.e., 30-60s).
2. You should also leave a whitespace between a word and its following citation: word (citation) rather than word(citation).
3. For TVA metrics, why do you compare the generated caption against the raw caption? Current SOTA methods for image–text and video–text alignment employ VQA-based metrics (Wiles et al., ICLR’25).
4. It would also be great (yet optional) if you could evaluate a SOTA model, as examples in Appendix E show high aesthetic scores for videos that are clearly much worse than SOTA, casting doubts on whether the proposed metrics will be able to capture differences in future models.

---
Wiles et al. Revisiting Text-to-Image Evaluation with Gecko: On metrics, prompts, and human ratings. ICLR’25

---

> ### Author Response · Authors · 2025-12-02
> **Response to Reviewer f6iZ**
>
> We sincerely appreciate your thoughtful feedback and the time you invested in reviewing our work. Below, we provide detailed responses to your comments.

---

> ### Author Response · Authors · 2025-12-02
> **Response to Reviewer f6iZ**
>
> > **W1**: For a paper presenting a new dataset, none of the actual prompts (and corresponding real videos) are shown in the main paper. This makes it difficult to understand the data creation and the complexity of the prompts from the main paper alone. I only saw 3 prompts in Appendix E.
>
> > **W2**: Details and takeaways from the prompt generation are missing. It would be informative to know how each of the stages (raw captioning, self-refine, manual review) change the prompt set, so the community can decide whether and how to build on top of this pipeline.
>
> **R1**: Thanks for your suggestions about this part. In fact, we failed to include any actual prompts for our evaluation in the main body primarily due to the page limit. However, to address your concern, we would like to provide at least one instances and illustrate each stage during our prompt construction process in Figure 3 to make our prompt creation more apparent. Please see it in our new submission.
>
> > **W3**: A large portion of the paper is used to introduce a large number of metrics to measure different aspects of generation. Nevertheless, some of the metrics are not clearly defined, entirely referring the reader to other papers (eg. L281). While details can be read in other papers, the main body of this paper should be self-contained, providing the minimum details needed for the reader to understand the methodology.
>
> **R2**: Thank you for the suggestion. Our intention was not to omit metric definitions simply because they appear in prior work, but to keep the main manuscript concise and focused on our core contributions. We were concerned that lengthy descriptions of previously established metrics might dilute the clarity and novelty of the paper. To ensure the paper remains self-contained, we have now added our own detailed explanations of all relevant metrics in the Appendix B.9 and included clear references to them in the main text. Please see it in our new submission.
>
> > **W5**: The number of metrics is also substantial, making it hard to process the different results. While more metrics can capture more nuances, it is important for a benchmark to provide a clear (sub)set of metrics to drive progress in the area.
>
> **R3**: We appreciate the reviewer’s concern regarding the large number of metrics. We will address this question as follows:
>
> 1. **Rationale for a Multi-Metric Design**: Long video generation inherently involves multiple dimensions—such as multi-scene transitions, subject consistency, event coherence, and storyline alignment, which cannot be reliably captured by a small set of metrics alone. This motivates the breadth of our evaluation dimensions.
> 2. **Ensuring Interpretability and Practical Utility**: We fully agree that a benchmark should remain interpretable and drive progress. To this end, our metrics are structured hierarchically into five major dimensions with clear semantic grouping. Each sub-metric targets a distinct and essential aspect of long-video quality, ensuring clarity rather than redundancy.
> 3. **Consistency with Established Benchmarking Practices**:  Our design follows the same philosophy adopted in prior video-generation benchmarks such as *VBench* [1] and *EvalCrafter* [2], both of which employ diverse sets of metrics to capture various facets of video quality, even though they focus on shorter videos. These works demonstrate that a broad and multidimensional evaluation protocol is both reasonable and necessary for comprehensive video assessment.
>
> > **W8**: It would also be great to measure how the real, ground-truth videos fare with respect to the metrics. This could be seen as some kind of upper-bound, which would increase credibility in the results.
>
> **R4**: We appreciate the reviewer’s insightful suggestion and have included the corresponding results in the rebuttal version. However, we would like to clarify that the ground-truth videos do not necessarily constitute an upper bound under our benchmark. In our setup, the ground-truth videos primarily serve as the basis for constructing prompts rather than targets to which generated videos must closely adhere. Even when we leverage powerful MLLMs/LLMs to create and enrich prompts (including incorporating HERD-derived descriptions of the ground-truth videos), this does not imply that the generated videos are expected to closely resemble the ground-truth content. Under our evaluation protocol, a generated video can still achieve high scores as long as it is coherent, well-aligned with the prompt, and of high quality—even if it differs substantially from the original ground-truth video.

---

> ### Author Response · Authors · 2025-12-02
> **Response to Reviewer f6iZ**
>
> > **W9**: Some older yet relevant related work is missing. This includes models such as Phenaki (Villegas et al., ICLR’23), which was the first model for multi-event generation, and benchmarks such as StoryBench (Bugliarello et al., NeurIPS’23) which evaluates multi-prompt generations and shows that metrics like DOVER do not correlate well with human preferences.
>
> **R5**: Thank you for pointing out these citation omissions. We’ve complemented these citation into our related work and stress them in blue font. Please see it in our new submission.
>
> > **Q1**: It would be nice to define early on in the paper what “long” video generation means (i.e., 30-60s).
>
> **R6**: We appreciate the reviewer for catching this issue and we've add obvious explanation about this definition in our Introduction and stress it in blue font. Please see it in our new submission.
>
> > **Q2**: You should also leave a whitespace between a word and its following citation: word (citation) rather than word(citation).
>
> **R7**: We thank the reviewer for carefully pointing out this fundamental error caused by our hasty submission and careless inspection. We have thoroughly checked the citation parts of the entire text and corrected all the related errors. Please see it in our new submission.
>
> > **Q3**: For TVA metrics, why do you compare the generated caption against the raw caption? Current SOTA methods for image–text and video–text alignment employ VQA-based metrics (Wiles et al., ICLR’25).
>
> **R8**: We appreciate the reviewer’s suggestion regarding VQA-based alignment methods for evaluating video-text alignment. Our choice of comparing generated captions with the raw prompt caption is motivated by several considerations:
>
> 1. **Preserving Global and Relational Information**: VQA-based evaluations typically decompose a caption into independent QA pairs. While effective for verifying isolated facts, this decomposition often loses the relational structure inherent in long-video descriptions—such as temporal dependencies or causal chains between events. For instance, even if answers to “Did event A occur?” and “Did event B occur?” are both correct, the model may still fail to preserve their intended order. Capturing such higher-order relations would require designing a large set of complex, structured questions, which remains nontrivial for long and detailed prompts.
> 2. **Stability and Robustness Considerations**: VQA results are sensitive to question formulation and susceptible to hallucination from the underlying VLM. In contrast, video captioning is a core capability extensively covered in pretraining, making caption-based alignment more stable and reflective of the full content of the generated video. Empirically, we observe that captioning provides more reliable semantic coverage for long-form content than current VQA pipelines.
> 3. **Practicality and Lack of Suitable Long-Video VQA Frameworks**: Applying VQA to our setting would require transforming each long and complex prompt into a large battery of queries, along with carefully engineered structures to model entity interactions and temporal relationships—requirements that go significantly beyond the image–text setting discussed in your mentioned work. To the best of our knowledge, no existing VQA-based video–text alignment method currently supports long-video scenarios with rich, multi-entity, multi-event prompts. Therefore, developing a robust VQA-based alternative would require substantial methodological innovation, which is beyond the focus of our benchmark.

---

> ### Author Response · Authors · 2025-12-02
> **Response to Reviewer f6iZ**
>
> > **Q4**: It would also be great (yet optional) if you could evaluate a SOTA model, as examples in Appendix E show high aesthetic scores for videos that are clearly much worse than SOTA, casting doubts on whether the proposed metrics will be able to capture differences in future models.
>
> **R9**: We really agree with you to some extent. But in our opinions, the following issues are supposed to take into consideration:
>
> 1. Even though future models may be quite powerful, we regard our proposed LoCoT2V-Bench as the first attempt to comprehensively evaluate the performance of existing LVG methods rather than a long-lasting challenging benchmark. It allows for subsequent refinement to cope with more complex and fine-grained occasions in the future. Similar work like *VBench* [1] also experiences multi-round refinement to adapt to more occasions such as longer videos (*VBench++* [3]) and videos with complex plot (*VBench 2.0* [4]). In summary, we believe that evaluation and methodology are not isolated, but rather develop in synergy and we would like refine LoCoT2V-Bench if needed in the future.
> 2. In fact, for long video generation include multi-scene video generation, there still lacks a unified and reliable benchmark for the video quality evaluation. A large amount of previous works mainly demonstrate their effect through case study. Therefore, it's hard to decide which one is the SOTA method. What's worse, most of commercial API could only support T2V generation for videos that are 5-10 seconds with limited content and we failed to find one that could serve as SOTA. While we were unable to identify any proprietary text-to-video system that can directly generate 30–60s multi-scene videos required by our benchmark, we explored an alternative way to include closed-source models. Specifically, we tested a pipeline that combines storyboard-based keyframe generation with closed-source Image-to-Video models, where we use *story-adapter* [5] with proprietary I2V model *seedance-pro-1.0* [6]. Due to the high computational and financial cost of these proprietary services, we evaluated only three examples and included the results as case studies in the appendix. The updated manuscript now contains these additional comparisons.
>
> ---
>
> [1] Huang Z et al. Vbench: Comprehensive benchmark suite for video generative models. CVPR'24
>
> [2] Liu Y et al. Evalcrafter: Benchmarking and evaluating large video generation models. CVPR'24
>
> [3] Huang Z et al. Vbench++: Comprehensive and versatile benchmark suite for video generative models. arXiv:2411.13503
>
> [4] Zheng D et al. Vbench-2.0: Advancing video generation benchmark suite for intrinsic faithfulness. arXiv:2503.21755
>
> [5] Mao J et al. Story-Adapter: A Training-free Iterative Framework for Long Story Visualization. arXiv:2410.06244
>
> [6] Gao Y et al. Seedance 1.0: Exploring the Boundaries of Video Generation Models. arXiv:2506.09113

---

### Author Response · Authors · 2025-12-02
**General Response to Reviewers for Human Verification Experiments**

We sincerely thank all reviewers for highlighting the importance to assess the alignment between our evaluation and human verification.
We fully agree that this direction would further strengthen the contribution of our work. However, due to the limited time window of the rebuttal period and the substantial engineering/manpower effort required to properly address this issue, we are unfortunately unable to complete a full investigation within the scope of the rebuttal. That said, we want to emphasize that the core conclusions of our paper remain valid under the current experimental setup, and the requested extension, while valuable, is orthogonal to our main claims. We genuinely appreciate the reviewers’ insightful suggestion, and we will include this extension as a key component of our future work, where we will have sufficient time and resources to conduct a comprehensive evaluation.

---

### Meta-Review · Area_Chair_onsr · 2026-01-06

**Summary:**

AC agrees with some of the points the reviewers have brought up. The metrics seem to be unclear, and a lot of details are missing, making it hard to judge the quality of the datasets.

Moreover without human assessment, it is also hard to pass this as a bona fide "long video" datasets. The writing needs to be improved somehow. A reviewer questioned "what is a long video" -- this has not been properly handled in the paper.

In general, AC thinks that this work should be refined and send to the next venue.

**Reviewer Concerns:**

Some of the missing details are provided in the rebuttal but most are all addressed.

**Reviewer Scores:**

AC doubts the scores will change.

---

### Decision · Program_Chairs · 2026-01-26

Reject